# Cooperative Distribution Alignment
# via JSD Upper Bound

**Wonwoong Cho**[*]
Purdue University[†]
cho436@purdue.edu

**Ziyu Gong**[*]
Purdue University[†]
gong123@purdue.edu

**David I. Inouye**
Purdue University[†]
dinouye@purdue.edu

## Abstract

Unsupervised distribution alignment estimates a transformation that maps two or more source distributions to a shared aligned distribution given only samples from each distribution. This task has many applications including generative modeling, unsupervised domain adaptation, and socially aware learning. Most prior works use adversarial learning (i.e., min-max optimization), which can be challenging to optimize and evaluate. A few recent works explore non-adversarial flow-based (i.e., invertible) approaches, but they lack a unified perspective and are limited in efficiently aligning multiple distributions. Therefore, we propose to unify and generalize previous flow-based approaches under a single non-adversarial framework, which we prove is equivalent to minimizing an upper bound on the Jensen-Shannon Divergence (JSD). Importantly, our problem reduces to a min-min, i.e., cooperative, problem and can provide a natural evaluation metric for unsupervised distribution alignment. We show empirical results on both simulated and real-world datasets to demonstrate the benefits of our approach. Code is available at `https://github.com/inouye-lab/alignment-upper-bound`.

## 1 Introduction

In many cases, a practitioner has access to multiple related but distinct distributions such as agricultural measurements from two farms, experimental data collected in different months, or sales data before and after a major event. *Unsupervised* distribution alignment (UDA) is the ML task aimed at aligning these related but distinct distributions in a shared space, *without* any pairing information between the samples from these distrbibutions (i.e., unsupervised). This task has many applications such as generative modeling (e.g., [1]), unsupervised domain adaptation (e.g., [2, 3]), batch effect mitigation in biology (e.g., [4]), and fairness-aware learning (e.g., [5]).

The most common approach for obtaining such alignment transformations stems from Generative Adversarial Networks (GAN) [6], which can be viewed as minimizing a *lower bound* on the Jensen-Shannon Divergence (JSD) between real and generated distributions. The lower bound is tight if and only if the inner maximization is solved perfectly. CycleGAN [1] maps between *two* datasets via two GAN objectives between the domains and a cycle consistency loss, which encourages approximate invertibility of the transformations.

However, adversarial learning can be challenging to optimize in practice (see e.g. [7–11]) in part because of the competitive nature of the min-max optimization problem. Perhaps more importantly, the research community has resorted to surrogate evaluation metrics for GAN because likelihood computation is intractable. Specifically, the commonly accepted Frechet Inception Distance (FID) [12] is only applicable to image or auditory data for which there exist publicly available classifiers

---

[*]Equal contributions

[†]Elmore Family School of Electrical and Computer Engineering, Purdue University, West Lafayette, IN

trained on large-scale data. Moreover, the concrete implementation of FID can have issues due to seemingly trivial changes in image resizing algorithms [13].

Recently, flow-based methods with a tractable likelihood have been proposed for the UDA task [2, 14–16]. Specifically, iterative flow methods [14, 15] proposed alternative approaches to distribution alignment via iteratively building up a deep model via simpler maps. AlignFlow [2] leverages invertible models to make the model cycle-consistent (i.e., invertible) *by construction* and introduces exact log-likelihood loss terms derived from standard flow-based generative models that complement the adversarial loss terms. On the other hand, log-likelihood ratio minimizing flows (LRMF) [16] use invertible flow models and density estimation for distribution alignment without adversarial learning and define a new metric based on the log-likelihood ratio.

However, iterative flow models [14, 15] do not explicitly reduce a global divergence measure and actually solve non-standard adversarial problems via alternating optimization (see [15]). AlignFlow [2] assumes that the shared density model for all distributions is a fixed Gaussian and lacks an explicit alignment metric. LRMF [16] may only partially align distributions if the target distribution is not in the model class. Also, the LRMF metric is limited because it is only defined for two distributions and depends on the shared density model class.

To address these issues, we unify existing flow-based methods (both AlignFlow and LRMF) under a common cooperative (i.e., non-adversarial) framework by proving that a minimization over a shared density model is a variational upper bound of the JSD. The unifying theory also suggests a natural domain-agnostic metric for UDA that can be applied to any domain including tabular data (where FID is inapplicable). This metric is analogous to the Evidence Lower Bound (ELBO), i.e., it is a variational bound that is useful for both training models and comparing models via held-out test evaluation. Furthermore, this unification enables straightforward and parameter-efficient multi-distribution alignment because the distributions share a latent space density model.

We summarize our contributions as follows:

- We prove that a minimization over a shared variational density model is a *variational upper bound* on a generalized version of JSD that allows for more than two distributions. Importantly, we theoretically quantify the bound gap and show that it can be made tight if the density model class is flexible enough.

- Based on this JSD upper bound, we derive a novel unified framework for cooperative (i.e., non-adversarial) flow-based UDA that includes a novel domain-agnostic AUB metric and explain its relationship to prior flow-based alignment methods.

- Throughout experiments, we demonstrate that our framework consistently shows superior performance in both our proposed and existing measures on simulated and real-world datasets. We also empirically show that our model is more parameter-efficient than the baseline models.

**Notation**  We will denote distributions as $P_X(\boldsymbol{x})$ where $X$ is the corresponding random variable. Invertible functions will be denoted by $T(\cdot)$. We will use $X_j \sim P_{X_j}$ to denote the observed random variable from the $j$-th distribution. We will use $Z_j \triangleq T_j(X_j) \sim P_{Z_j} \equiv P_{T_j(X_j)}$ to denote the latent random variable of the $j$-th distribution after applying $T_j$ to $X_j$ (and note that $X_j = T_j^{-1}(Z_j)$). We will denote the mixtures of these observed or latent distributions as $P_{X_{\mathrm{mix}}} \triangleq \sum_j w_j P_{X_j}$ and $P_{Z_{\mathrm{mix}}} \triangleq \sum_j w_j P_{Z_j}$, where $\boldsymbol{w}$ is a probability vector. We denote KL divergence, entropy, and cross entropy as $\mathrm{KL}(\cdot, \cdot)$, $\mathrm{H}(\cdot)$, and $\mathrm{H_c}(\cdot, \cdot)$, respectively, where $\mathrm{KL}(P, Q) = \mathrm{H_c}(P, Q) - \mathrm{H}(P)$.

## 2   Alignment Upper Bound Loss

In this section, we will present our main theoretical result by proving an upper bound on the generalized JSD divergence, deriving our loss function based on this upper bound, and then showing that minimizing this upper bound results in aligned distributions assuming the model components have a large enough capacity.

**Background: Normalizing Flows and Invertible Models.**   Normalizing flows are generative models that have tractable distributions where exact likelihood evaluation and exact sampling are

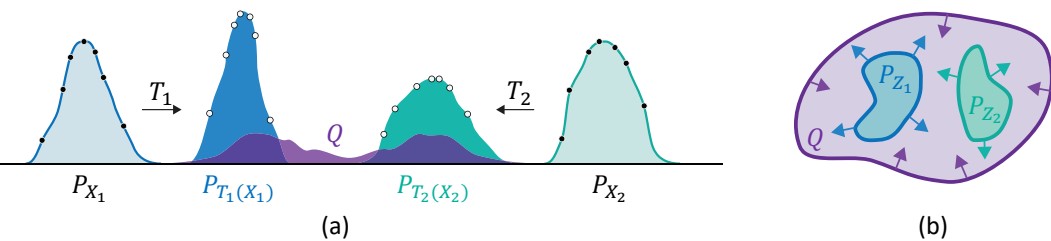

Figure 1: In our AUB framework, domain specific transformation functions $\{T_j(\cdot)\}_{j=1}^k$ and a density model $Q$ are cooperatively trained to make the transformed representations be indistinguishable in the shared latent space, i.e., aligned. **(a)** 1-D example. By minimizing our proposed AUB loss, the transformation functions $T_1$ and $T_2$ are trained to map the corresponding distributions $P_{X_1}$ and $P_{X_2}$ to latent distributions $P_{T_1(X_1)}$ and $P_{T_2(X_2)}$ that have higher likelihood with respect to a base distribution $Q$. The density model $Q$, on the other hand, is trained to fit the mixture of the latent distributions $P_{T_1(X_1)}$ and $P_{T_2(x_2)}$. **(b)** Intuitively, the optimization process of our method can be viewed as making the $Q$ distribution tight around the mixture of latent distributions to increase the likelihood (i.e., MLE) while the transformation functions $T_1$ and $T_2$ are encouraged to expand to fill the latent space defined by $Q$. Eventually, the latent distributions and $Q$ converge to the same distribution, which means that they are aligned.

possible [17]. Flow models leverage the change of variables formula to create an invertible mapping $T$ such that $P_X(\boldsymbol{x}) = P_Z(T(\boldsymbol{x}))|J_T(\boldsymbol{x})|$ where $P_Z$ is a known latent distribution and $|J_T(\boldsymbol{x})|$ is the absolute value of determinant of the Jacobian of the invertible map $T$. To sample from $P_X$, one needs to first sample from the latent distribution $P_Z$ and then apply the inverse transform $T^{-1}$. Therefore, the key challenge in designing invertible models is to have computationally efficient inverse evaluation for sampling and Jacobian determinant calculation for training. Many approaches have been proposed by parameterizing mapping function $T$ as deep neural networks including autoregressive structures [18, 19], coupling layers [20, 21], ordinary differential equations[22], and invertible residual networks[23, 24]. Flow models can be then learned efficiently by maximizing the likelihood of the given data.

**Background: Generalized JSD.** We remind the reader of the generalized Jensen-Shannon divergence for more than two distributions, where the standard JSD is recovered if $w_1 = w_2 = 0.5$.
**Definition 2.1** (Generalized Jensen-Shannon Divergence (GJSD) [25]). Given $k$ distributions $\{P_{Z_j}\}_{j=1}^k$ and a corresponding probability weight vector $\boldsymbol{w}$, the generalized Jensen-Shannon divergence is defined as (proof of equivalence in **??**.):

$$\text{GJSD}_{\boldsymbol{w}}(P_{Z_1}, \cdots, P_{Z_k}) \triangleq \sum_j w_j \, \text{KL}(P_{Z_j}, \sum_j w_j P_{Z_j}) \equiv \text{H}\left(\sum_j w_j P_{Z_j}\right) - \sum_j w_j \, \text{H}(P_{Z_j}).$$

## 2.1 GJSD Variational Upper Bound

The goal of distribution alignment is to find a set of transformations $\{T_j(\cdot)\}_{j=1}^k$ (which will be invertible in our case) such that the latent distributions align, i.e., $P_{T_j(X_j)} = P_{T_{j'}(X_{j'})}$ or equivalently $P_{Z_j} = P_{Z_{j'}}$ for all $j \neq j'$. Given the properties of divergences, this alignment will happen if and only if $\text{GJSD}(P_{Z_1}, \cdots, P_{Z_k}) = 0$. Thus, ideally, we would minimize GJSD directly, i.e.,

$$\min_{T_1,\cdots,T_k \in \mathcal{T}} \text{GJSD}(P_{T_1(X_1)}, \cdots, P_{T_k(X_k)}) \equiv \min_{T_1,\cdots,T_k \in \mathcal{T}} \text{H}\left(\sum_j w_j P_{T_j(X_j)}\right) - \sum_j w_j \, \text{H}(P_{T_j(X_j)}), \tag{1}$$

where $\mathcal{T}$ is a class of invertible functions. However, we cannot evaluate the entropy terms in Eqn. 1 because we do not know the density of $P_{X_j}$; we only have samples from $P_{X_j}$. Therefore, we will upper bound the first entropy term in Eqn. 1 ($\text{H}(\sum_j w_j P_{T_j(X_j)})$) using a variational density model and decompose the other entropy terms via the change of variables formula for invertible functions.

**Theorem 2.2** (GJSD Variational Upper Bound). *Given an variational density model class $\mathcal{Q}$, we form a GJSD variational upper bound:*

$$\text{GJSD}_{\boldsymbol{w}}(P_{Z_1}, \cdots, P_{Z_k}) \leq \min_{Q \in \mathcal{Q}} \text{H}_c(P_{Z_{\text{mix}}}, Q) - \sum_j w_j \, \text{H}(P_{Z_j}),$$

*where $P_{Z_{\text{mix}}} \triangleq \sum_j w_j P_{Z_j}$ and the bound gap is exactly $\min_{Q \in \mathcal{Q}} \text{KL}(P_{Z_{\text{mix}}}, Q)$.*

*Proof of Theorem 2.2.* For any $Q \in \mathcal{Q}$, we have the following upper bound:

$$\text{GJSD}_{\boldsymbol{w}}(P_{Z_1}, \cdots, P_{Z_k}) = \underbrace{\text{H}_c(P_{Z_{\text{mix}}}, Q) - \text{H}_c(P_{Z_{\text{mix}}}, Q)}_{=0} + \text{H}(P_{Z_{\text{mix}}}) - \sum_j w_j \text{H}(P_{Z_j})$$

$$= \text{H}_c(P_{Z_{\text{mix}}}, Q) - \text{KL}(P_{Z_{\text{mix}}}, Q) - \sum_j w_j \text{H}(P_{Z_j})$$

$$\leq \text{H}_c(P_{Z_{\text{mix}}}, Q) - \sum_j w_j \text{H}(P_{Z_j}),$$

where the first equals is merely inflating by $\text{H}_c(P_{Z_{\text{mix}}}, Q)$, the inequality is by the fact that KL divergence is non-negative, and the bound gap is equal to $\text{KL}(P_{Z_{\text{mix}}}, Q)$. The $Q$ that achieves the minimum in the upper bound is equivalent to the $Q$ that minimizes the bound gap, i.e., $Q^* = \arg\min_{Q \in \mathcal{Q}} \text{H}_c(P_{Z_{\text{mix}}}, Q) - \sum_j w_j \text{H}(P_{Z_j}) = \arg\min_{Q \in \mathcal{Q}} \text{H}_c(P_{Z_{\text{mix}}}, Q) - \text{H}(P_{Z_{\text{mix}}}) = \arg\min_{Q \in \mathcal{Q}} \text{KL}(P_{Z_{\text{mix}}}, Q)$, where the second equality is because the entropy terms are constant with respect to $Q$ and the last is by the definition of KL divergence. $\square$

The tightness of the bound depends on how well the class of density models $\mathcal{Q}$ (e.g., mixture models, normalizing flows, or autoregressive densities) can approximate $P_{Z_{\text{mix}}}$; notably, the bound can be made tight if $P_{Z_{\text{mix}}} \in \mathcal{Q}$. Also, one key feature of this upper bound is that the cross entropy term can be evaluated using only samples from $P_{X_j}$ and the transformations $T_j$, i.e., $\text{H}_c(P_{Z_{\text{mix}}}, Q) = \sum_j w_j \mathbb{E}_{P_{X_j}}[-\log Q(T_j(\boldsymbol{x}_j))]$. However, we still cannot evaluate the other entropy terms $\text{H}(P_{Z_j})$ since we do not know the density functions of $P_{Z_j}$ (or $P_{X_j}$). Thus, we leverage the fact that the $T_j$ functions are invertible to define an entropy change of variables.

**Lemma 2.3** (Entropy Change of Variables). *Let $X \sim P_X$ and $Z \triangleq T(X) \sim P_Z$, where $T$ is an invertible transformation. The entropy of $Z$ can be decomposed as follows:*

$$\text{H}(P_Z) = \text{H}(P_X) + \mathbb{E}_{P_X}[\log|J_T(\boldsymbol{x})|], \tag{2}$$

*where $|J_T(\boldsymbol{x})|$ is the absolute value of the determinant of the Jacobian of $T$.*

The key insight from this lemma is that $\text{H}(P_X)$ is a constant with respect to $T$ and can thus be ignored when optimizing $T$, while $\mathbb{E}_{P_X}[\log|J_T(\boldsymbol{x})|]$ can be approximated using only samples from $P_X$ (formal proof in **??**).

## 2.2 Alignment Upper Bound (AUB)

Combining Theorem 2.2 and Lemma 2.3, we can arrive at our final objective function which is equivalent to minimizing the variational upper bound on the GJSD:

$$\text{GJSD}_{\boldsymbol{w}}(P_{Z_1}, \cdots, P_{Z_k}) \leq \min_{Q \in \mathcal{Q}} \text{H}_c(P_{Z_{\text{mix}}}, Q) - \sum_j w_j \text{H}(P_{Z_j}) \tag{3}$$

$$= \min_{Q \in \mathcal{Q}} \sum_j w_j \mathbb{E}_{P_{X_j}}[-\log Q(T_j(\boldsymbol{x}))|J_{T_j}(\boldsymbol{x})|] - \sum_j w_j \text{H}(P_{X_j}), \tag{4}$$

where the cross entropy term is replaced by its definition provided above, and the last term $-\sum_j w_j \text{H}(P_{X_j})$ is constant with respect to $T_j$ functions so they can be ignored during optimization. We formally define this loss function as follows.

**Definition 2.4** (Alignment Upper Bound Loss). *Given $k$ continuous distributions $\{P_{X_j}\}_{j=1}^k$, a class of continuous distributions $\mathcal{Q}$, and a probability weight vector $\boldsymbol{w}$, the alignment upper bound loss is defined as follows:*

$$\mathcal{L}_{\text{AUB}}(T_1, \cdots, T_k; \{P_{X_j}\}_{j=1}^k, \mathcal{Q}, \boldsymbol{w}) \triangleq \min_{Q \in \mathcal{Q}} \sum_j w_j \mathbb{E}_{P_{X_j}}[-\log|J_{T_j}(\boldsymbol{x})| Q(T_j(\boldsymbol{x}))], \tag{5}$$

*where $T_j$ are invertible and $|J_{T_j}(\boldsymbol{x})|$ is the absolute value of the Jacobian determinant.*

Notice that this alignment loss can be seen as learning the best base distribution given fixed flow models $T_j$. We now consider the theoretical optimum if we optimize over all invertible functions.

**Theorem 2.5** (Alignment at Global Minimum of $\mathcal{L}_{\text{AUB}}$). *If $\mathcal{L}_{\text{AUB}}$ is minimized over the class of all invertible functions, a global minimum of $\mathcal{L}_{\text{AUB}}$ implies that the latent distributions are aligned, i.e., $P_{T_j(X_j)} = P_{T_{j'}(X_{j'})}$ for all $j \neq j'$. Notably, this result holds regardless of $\mathcal{Q}$.*

Informally, this can be proved by showing that the problem decouples into separate normalizing flow losses where $Q$ is the base distribution and the optimum is achieved only if $P_{T_j(X_j)} = Q$ for all $T_j$ (formal proof in **??**). This alignment of the latent distributions also implies the translation between any of the *observed* component distributions. The proof follows directly from Theorem 2.5 and the change of variables formula.

**Corollary 2.6** (Translation at Global Minimum of $\mathcal{L}_{\text{AUB}}$). *Similar to Theorem 2.5, a global minimum of $\mathcal{L}_{\text{AUB}}$ implies translation between any component distributions using the inverses of $T_j$, i.e., $P_{T_{j'}^{-1}(T_j(X_j))} = P_{X_{j'}}$ for all $j \neq j'$.*

As seen in Alg. 1, we use a simple alternating optimization scheme for training our translation models and variational distribution with cooperative (i.e., min-min) AUB objective, i.e., we aim to optimize:

$$\min_{T_1, \cdots, T_k \in \mathcal{T}} \min_{Q \in \mathcal{Q}} \sum_j w_j \mathbb{E}_{P_{X_j}} \left[ -\log |J_{T_j}(\boldsymbol{x})| \, Q(T_j(\boldsymbol{x})) \right]. \tag{6}$$

We emphasize that our framework allows *any* invertible function for $T_j$ (e.g., coupling-based flows [20], neural ODE flows [22], or residual flows [23]) and *any* density model class for $Q_z$ (e.g., kernel densities in low dimensions, mixture models, autoregressive densities [26], or normalizing flows [21]). Even VAEs [27] could be used where the log likelihood term is upper bounded by the negative ELBO, which will ensure the objective is still an upper bound of GJSD.

---

**Algorithm 1** Training algorithm for AUB

---

**Input:** Datasets $\{X_j\}_{j=1}^k$ for $k$ domains; $n$ as the batch size; $x_j$ as a minibatch for the $j$-th domain; normalizing flow models $\{T_j(x_j; \theta_j)\}_{j=1}^k$; density model $Q(z; \phi)$; learning rate $\eta$; maximum epoch $E_{max}$
**Output:** $\{\hat{\theta}\}_{j=1}^k$;
**for** epoch $= 1, E_{max}$ **do**
    **for** each batch $\{x_j\}_{j=1}^k$ **do**
        $\phi \leftarrow \phi + \eta \nabla_\phi \frac{1}{k} \sum_{j=1}^k \frac{1}{n} \sum_{i=1}^n \log Q(T_j(x_{i,j}; \theta_j); \phi)$
    **end for**
    **for** each batch $\{x_j\}_{j=1}^k$ **do**
        $\forall j, \quad \theta_j \leftarrow \theta_j + \eta \nabla_{\theta_j} \frac{1}{n} \sum_{i=1}^n \log |J_{T_j}(x_{i,j}; \theta_j)| \, Q(T_j(x_{i,j}; \theta_j); \phi)$
    **end for**
**end for**

---

**AUB for UDA is like ELBO for density estimation.** Although AUB and ELBO are for fundamentally different tasks, we would like to point out the similarities between AUB and ELBO. First, both are *variational* bounds of the quantity of interest where the tightness of the bounds depend on the optimization of the variational distributions. Second, both can be made tight if the class of variational distributions is powerful enough. Third, both can be used to train a model by minimizing the objective on training data. Fourth, while neither can be used as an absolute performance metric, they can both be used to evaluate the *relative* performance of models on held-out test data. Thus, AUB can be used for UDA as ELBO has been used for density estimation with the same strengths and weaknesses such as being a relative metric and requiring an auxiliary model for evaluation.

## 3 Relationship to Prior Works

**AlignFlow without adversarial terms is a special case.** As illustrated in Fig. 2, AlignFlow [2] *without* adversarial loss terms is a special case of our method for two distributions where the density model class $\mathcal{Q}$ only contains the standard normal distribution (i.e., a singleton class). Thus, AlignFlow can be viewed as initially optimizing a poor upper bound on JSD; however, the JSD bound becomes tighter as training progresses because the latent distributions *independently* move towards the same normal distribution. By using the same architecture of $T$ and $Q$, AlignFlow can be also viewed as sharing the last few layers of the $T$'s; however, we note that our approach allows for $Q$ that are not flows, e.g., autoregressive densities or mixture models as in our toy experiments that even have alternative non-SGD learning algorithms.

**LRMF is special case with only one transformation.** As illustrated in Fig. 2, Log-likelihood ratio minimizing flows (LRMF) [16] is also a special case of our method for only two distributions, where one transformation is fixed at the identity (i.e., $T_2 = \text{Id}$). While the final LRMF objective is a special case of ours, the theory is developed from a different but complementary perspective. The LRMF metric depends on the shared density model class, which enables a zero point (or absolute value) of the metric to be estimated but requires fitting extra density models. Usman et al. [16] do not uncover

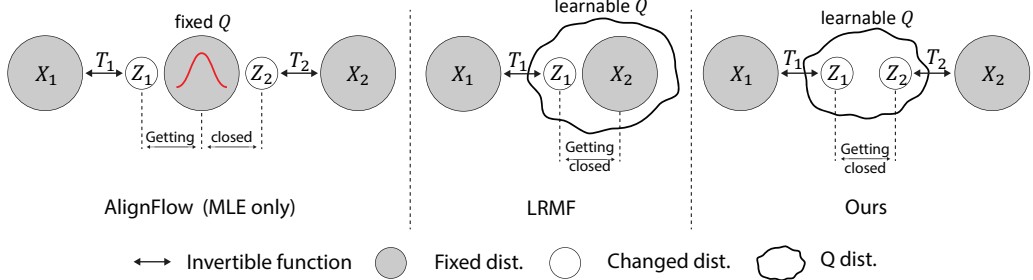

Figure 2: These illustrations of flow-based alignment methods AlignFlow, LRMF, and AUB demonstrate the differences between each setup. Transformation functions in AlignFlow are independently trained to map the distributions to a fixed standard normal distribution. $T_1$ in LRMF is trained to directly map the source distribution $P_{X_1}$ onto the target distribution $P_{X_2}$, i.e., $T_2$ is the identity. The density model $Q$ in LRMF is not fixed and is trained to fit the mixture $\frac{1}{2}P_{Z_1} + \frac{1}{2}P_{X_2}$. In our AUB setup, $T_1$ and $T_2$ are trained to map the source distributions $P_{X_1}$ and $P_{X_2}$ onto the shared $Q$ distribution, while $Q$ is trained to fit the mixture $w_1 P_{Z_1} + w_2 P_{Z_2}$. In every setup, the latent distributions $Z_1$ and $Z_2$ move closer to the target distribution as training progresses. Details are provided in section 3.

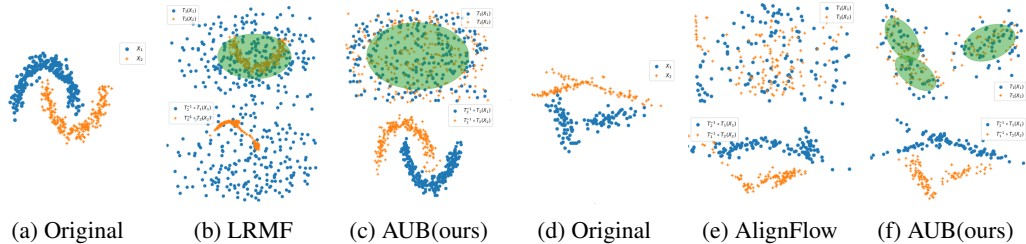

(a) Original      (b) LRMF      (c) AUB(ours)      (d) Original      (e) AlignFlow      (f) AUB(ours)

Figure 3: (a-c) LRMF, which only has *one* transformation $T$ may not be able to align the datasets if the density model class $\mathcal{Q}$ is not expressive enough (in this case Gaussian distributions), while using two transformations as in our framework can align them. (d-f) AlignFlow (without adversarial terms) may not align because $Q$ is fixed to a standard normal, while our approach with learnable mixture of Gaussians for $Q$ is able to learn an alignment (both use the same $T_j$ models). Top row is latent space and bottom is the data translated into the other space.

the connection of the objective as an upper bound on JSD regardless of the density model class[3]. Additionally, to ensure alignment, LRMF requires that the density model class includes the true target distribution because only one invertible transform is used, while our approach can theoretically align even if the shared density model class is weak. In other words, our bound holds regardless of the model class $\mathcal{Q}$, whereas the LRMF-JSD discussion further assumes that the target distribution needs to be in the $\mathcal{Q}$ model class (see Theorem 2.5 and our simulated experiments).

## 4 Experiment

We analyze the performance of our proposed framework comparing to the relevant flow-based baseline models. In subsection 4.1, we experiment on a toy dataset to clearly show the benefit of our method over the baseline flow-based models in a controlled environment. In subsection 4.2 and subsection 4.3, we demonstrate AUB's superiority over baseline flow-based models on real-world datasets including tabular data and high-dimensional MNIST data. In subsection 4.4, we additionally conduct Domain Adaptation [2, 28] experiments for validating AUB alignment in the context of a downstream task. Implementation details are provided in **??**.

### 4.1 Toy dataset comparison with related works

**Single $T$ vs. Double $T$ (LRMF vs. Ours).** We first compare our method with LRMF [16] method. The task is to translate between the two half-circle distributions (i.e., "moons"). We compared

---

[3]LRMF did discuss a connection with JSD but only as "biased estimates of JSD", rather than a theoretic upper bound of JSD.

our AUB setup with two maps $T_1$ and $T_2$ to the LRMF setup with only $T_1$ where $\mathcal{Q}$ is the set of independent Gaussian distributions. As illustrated in Fig. 3, the LRMF method fails to transform between $X_1$ and $X_2$. Even though $Q$ can model the transformed data $T_1(X_1)$, a Gaussian-based $Q$ cannot fully model the half-circle distribution of $X_2$. Therefore, LRMF fails to transform between two distributions. However, in the AUB setup, both latent distributions of $T_1(X_1)$ and $T_2(X_2)$ can be modeled by the same Gaussian distribution $Q$ because of the flexibility in both transformations, which leads to better translation results in this illustrative example. In conclusion, the performance of LRMF is limited by the choice of density models $\mathcal{Q}$; if $\mathcal{Q}$ fails to model the target distribution, distribution alignment may not be achieved.

**Simple Fixed $Q$ vs. Learnable $Q$ (AlignFlow vs. Ours).**    Next we compare our AUB setup with the AlignFlow [2, 3] setup. The task is to translate between two random Gaussian mixture datasets (i.e., "blobs"). We compared our AUB setup where $\mathcal{Q}$ is a mixture of Gaussians with the AlignFlow setup where $\mathcal{Q}$ is a fixed standard normal distribution (the models for $T_1$ and $T_2$ are the same). As illustrated in Fig. 3, the AlignFlow method fails to transform between $X_1$ and $X_2$, because the transformed dataset $T_1(X_1)$ and $T_2(X_2)$ failed to reach the fixed standard normal distribution $Q$. However, in the AUB setup, the shared density model $Q$ adapted to the distributions of $T_1(X_1)$ and $T_2(X_2)$ to enable a tighter alignment bound and thus the translation results are better. In conclusion, the performance of the AlignFlow model is limited by the power of the invertible functions.

### 4.2   Unsupervised Distribution Alignment on Tabular Datasets

To showcase the application-agnostic AUB metric and the parameter efficiency of our AUB framework, we conduct two experiments on real-world tabular datasets. In both experiments, we used four UCI tabular datasets [29] (MINIBOONE, GAS, HEPMASS, and POWER), following the same preprocessing as the MAF paper [30]. Train, validation, and test sets are 80%, 10%, and 10% of the data respectively. Also, the experiments are measured by test AUB defined in Definition 2.4, where a lower AUB score indicates better performance (see end of section 2.2 for discussion on using test AUB for evaluation). We emphasize that there is no natural metric for evaluating GAN-based alignment methods on tabular datasets. Thus, these experiments demonstrate one of the key benefits of our proposed framework over a GAN-based approach.

Our first experiment was designed to compare alignment performance between our proposed method and the baseline methods. To separate each dataset into two distributions, we choose the last input feature from each dataset and discretize it based on whether it is higher or lower than the median value, which ensures the datasets are of equal size. Given the divided dataset, two transformation functions ($T$) are trained to align the distributions. For baselines, we use AlignFlow MLE, Adv. only, and hybrid versions and LRMF on top of the original implementation. Because AlignFlow Adv. and hybrid setups optimize over a mixed objective of AUB (special case) and adversarial losses, to be fair, we additionally fit an identity-initialized flow model $Q$ to the final $T$'s. We use the same $T$ and $Q$ models wherever possible across all methods (e.g., the same $\mathcal{Q}$ is used for LRMF and AUB but AlignFlow has a fixed $Q$).

As shown in Table 1, our method shows better performance compared to other methods across all datasets. In particular, because ours and LRMF (trained with learnable $Q$) outperform AlignFlow MLE (trained with fixed $Q$), we can see that learnable $Q$ plays an important role in distribution alignment. We also observe that ours shows better performance than LRMF, where the gap may come from aligning in the shared latent space (ours) rather than aligning in the original data space (LRMF). AlignFlow Adv. only and hybrid versions show worse performance than ours with a large gap, which implies our proposed cooperative training is competitive to adversarial methods in aligning distributions of tabular data.

The second experiment demonstrates alignment between 8 distributions and is designed to show that our proposed method can be more efficient in terms of model parameter compared to the baseline methods. For this multi-distribution experiment, we separate each dataset into 8 domains by choosing the last three features from a given dataset and dividing the dataset by the three medians, e.g., $(+++),(++-), ..., (---)$. The results are shown in Table 2 where RealNVP (5) and RealNVP (10) indicates the number of coupling layers used in the RealNVP architecture respectively. LRMF is excluded because LRMF is only designed for aligning two distributions.

Comparing AlignFlow RealNVP (5) and RealNVP (10), we can see that AlignFlow can achieve better alignment with more than double amount of parameters. However, as can be observed in a

Table 1: Our method outperforms baselines in terms of AUB score (in nats, lower is better) for the domain alignment task on four tabular datasets. Numbers below each dataset indicate the number of features for the dataset.

|  | MINIB (42) | GAS (7) | HEPM (20) | POW (5) |
|---|---|---|---|---|
| LRMF | 12.79 | -6.17 | 18.49 | -0.93 |
| AF (MLE) | 14.08 | -6.52 | 19.37 | -0.77 |
| AF (Adv. only) | 18.18 | -3.15 | 21.70 | -0.39 |
| AF (hybrid) | 19.49 | -3.76 | 21.42 | -0.43 |
| Ours | **12.11** | **-7.09** | **18.26** | **-1.19** |

Table 2: Our AUB method may be more parameter efficient than AlignFlow especially for the multi-distribution setting (8 distributions in this case) where the number of parameters (in millions) and AUB score are show below.

|  | # T | # Q | Total | AUB |
|---|---|---|---|---|
| AF (MLE) T: RealNVP (5) | 1.46 | 0 | 1.46 | 20.16 |
| AF (MLE) T: RealNVP (10) | 4.54 | 0 | 4.54 | 19.85 |
| Ours T: R.(5), Q: R.(10) | 1.46 | 0.57 | **2.03** | **18.82** |

comparison between AlignFlow with RealNVP (10) and ours, our proposed method can achieve similar performance with the baseline model but with less than half the parameters. These two results suggest that our model can be more parameter efficient than AlignFlow for multi-distribution alignment. We hypothesize that our approach may scale better with respect to the number of distributions because our $Q$ shares parameters across the distributions and can capture the similarities between distributions.

## 4.3 Unsupervised Distribution Alignment on MNIST Dataset

We perform an image translation task on MNIST dataset[4] [31] to demonstrate that our distribution alignment method can be applied to high-dimensional datasets and to validate our AUB metric in a case where FID is also available. We train ours and baseline models with the digit images of 0, 1, and 2, and compare the translated results both quantitatively and qualitatively. Specifically, we use RealNVP invertible models for all translation maps $T_j$, as well as the density model $Q$. Note that all methods are flow-based models and thus images translated back to the original domain are exactly the same, which implies exact cycle consistency.

As represented in Table 3, both of our approaches outperform the baseline models in terms of FID and AUB. The lower AUB score indicates our method seems to align the distributions better than baselines.[5] We believe this result comes from our model setup, i.e., a learnable shared density model and transformation to a shared latent space. Specifically, AlignFlow with a fixed standard normal distribution as their $Q$ obtains worse AUB because the $Q$ is not powerful enough to model the complex shared space trained from the real world dataset. On the other hand, LRMF shows the lack of stability when trained with the relatively simple models that we are using across all methods, i.e., RealNVP $T$ and RealNVP $Q$. We expect this is caused by the restrictions of only using one $T$ for translation without a shared latent space and the fact that the $Q$ distribution must be able to model the target distribution to ensure alignment.

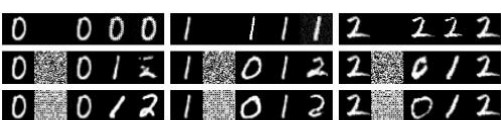

Figure 4: Qualitative translation results among MNIST digits 0-2 show that our method has better translation results than baselines where in each block the first column is the original digit, the second is the latent image, and the last three are translated results. Each row from the top indicates LRMF, AlignFlow, and ours respectively.

Table 3: This table of FID (top) and AUB (bottom) scores for the three pairwise MNIST translation tasks show that our method has overall better performance than baselines across both metrics. FID score for each translation task is calculated by averaging scores from each direction and AUB score is shown in nats. AF is short for AlignFlow.

|  | 0↔1 | 0↔2 | 1↔2 | Avg. |
|---|---|---|---|---|
| AF (MLE) | 38.90 | 71.17 | 61.33 | 57.13 |
| LRMF | 224.02 | 141.70 | 182.31 | 182.68 |
| Ours | **31.26** | **43.29** | **41.55** | **38.70** |
| AF (MLE) | -4797 | -4504 | -4834 | -4711 |
| LRMF | -713 | -592 | -1323 | -876 |
| Ours | **-4824** | **-4555** | **-4862** | **-4747** |

---

[4] under the terms of the Creative Commons Attribution-Share Alike 3.0 license

[5] While lower AUB scores may only mean a tighter upper bound, as with ELBO (see section 2.2, a lower upper bound generally means a better model.

The better quantitative performance of our methods can be corroborated by the qualitative results as seen in Fig. 4. AlignFlow shows less stable translation results than our method especially for translating to digit '2' from digits '0' and '1'. The second column of LRMF is set to be black because it does not have a latent representation, and LRMF fails to translate in this situation, which is why the translated results are nearly the same. We believe this phenomenon comes from the lack of expressivity of their $Q$ model. On the other hand, our model shows stable results across all translation cases, which are also quantitatively verified via the lower FID score. Note that our method has a shared space, so different transformation functions for each domain are trained together. This shared space may provide advantage in terms of sample complexity and computational complexity compared to multiple independent flows as in AlignFlow. Similarly, we conducted experiments on a ten-domain translation in **??** to illustrate that our method with a shared space can be easily scaled to more domain distributions with less number of parameters than the baseline model. Additionally, we visualize an interpolation over the shared space from the ten-domain experiment. Detailed descriptions and the results are provided in **??**

## 4.4 Domain Adaptation on USPS-MNIST dataset

We additionally conducted a domain adaptation (DA) experiment between MNIST to USPS for externally validating our alignment performance compared to baseline methods. We first reduce the dimensionality of both datasets to 32 using a pretrained variational autoencoder (VAE), which is trained jointly on both MNIST and USPS images without any label information (i.e., an unsupervised VAE). We then learn a translation maps in this latent space via our method and baselines. Finally, following the typical DA evaluation protocol, we train the classifier with source domain data and evaluate the performance by applying the classifier on the target domain data translated to the source domain. Image translation results can be obtained by forwarding the latent translated results to the pretrained decoder.

As seen in Table 4, our method (right column) performs better than baseline methods. This result is further empirical evidence that our novel cooperative training can be comparable to adversarial training in certain cases. As shown in Fig. 5, adversarial loss shows mode collapse, e.g., AF(Adv. only), AF(1e-2). without careful hyperparameter tuning. The other thing to note is AF (MLE) works for domain alignment itself (i.e., the translated result is in MNIST domain), but it does not maintain the class information (i.e., the digit changes from USPS to MNIST). We conjecture that this issue stems from AlignFlow's simple and fixed $Q$ distribution where arbitrary rotations of the latent space have equivalent likelihoods throughout the whole training process and thus class information is lost while transforming to and from such a $Q$ distribution. On the other hand, our learnable $Q$ guides the alignment training process such that class information is partially preserved during transformation. LRMF failed to align since the $Q$ distribution is not complex enough to model the density of the target distribution. This is the consistent result with our toy dataset experiment (Fig. 3, (a)-(c)). LRMF does not work well with the simple $Q$ while ours can align two different distributions with a simple $Q$ because of the shared space. Please note that all models in the table use approximately the same number of model parameters. Further comparisons are provided in **??**.

Table 4: Test classification accuracies for domain adaptation from USPS to MNIST. The number associated with AlignFlow (AF) is the coefficient of MLE term in its hybrid objective. Adversarial methods (i.e, AlignFlow Hybrid/AlignFlow Adv. only) are set to stop at 200 epochs.

|  | LRMF | AF(Adv.only) | AF(1e-2) | AF(1e-1) | AF(1e0) | AF(1e1) | AF(MLE) | AUB(ours) |
|---|---|---|---|---|---|---|---|---|
| Accuracy | 12.7% | 11.1% | 16.6% | 63.8% | 68.6% | 35.7% | 22% | **77.5%** |

## 5 Discussion

**Pros and cons of our method compared to adversarial method.** Flow-based methods have different benefits and limitations compared to adversarial methods for distribution alignment. As one clear difference, our framework provides an application-agnostic yet theoretically grounded evaluation metric for comparing alignment methods. Additionally, our min-min problem is fundamentally different than a min-max problem and avoids issues unique to min-max problems (as seen in Fig. 6). However, a key limitation of our approach compared to GANs is that it is restricted to invertible models. Also, our method requires training a density model, whereas GANs train a discriminator. Overall, we suggest our alignment approach is a feasible and fundamentally different alternative to adversarial, which is currently the only dominant approach. With our foundation, future work could

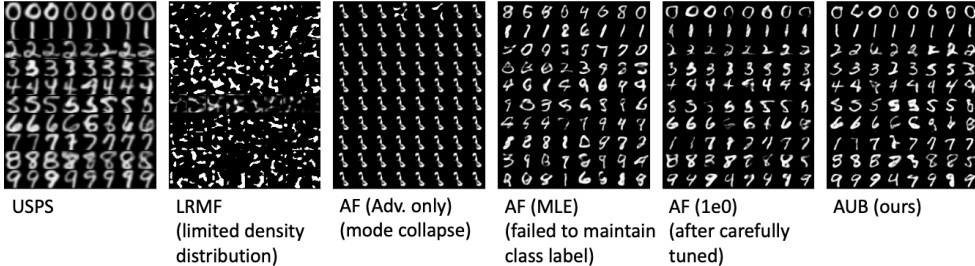

| USPS | LRMF (limited density distribution) | AF (Adv. only) (mode collapse) | AF (MLE) (failed to maintain class label) | AF (1e0) (after carefully tuned) | AUB (ours) |

Figure 5: Samples of translated images from USPS dataset. The leftmost image shows the original digits from USPS; all remaining images are the translated digits in MNIST domain.

focus on the performance aspects just as adversarial learning has been improving over the past seven years but is still an active area of research.

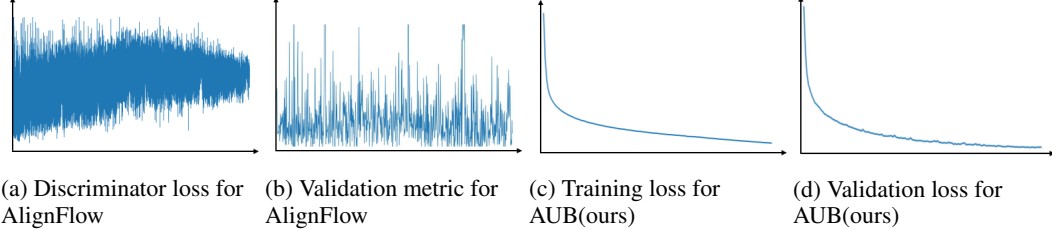

(a) Discriminator loss for AlignFlow  (b) Validation metric for AlignFlow  (c) Training loss for AUB(ours)  (d) Validation loss for AUB(ours)

Figure 6: The losses for AlignFlow with adversarial learning oscillate and are unstable during training, while our AUB losses show show smooth convergence because of cooperative (i.e., min-min) training.

**Alternating minimization against vanishing gradient problem.**   As revealed in previous works [16, 32], the vanishing gradient problem can occur while minimizing a JSD approximation. On the same line, AUB can also suffer from vanishing gradient if the distributions are disjoint, and $Q$ is fit very well. However, any $Q$ that does not match $P_{Z_{mix}}$ provides an upper bound that we conjecture is smoother than the true JSD. Thus, in experiments, we have found that alternating minimization does not suffer from vanishing gradients because $Q$ is not fully fitted at each step but provides a smooth upper bound. Yet, deeper theoretic and empirical analysis is needed in future work to fully understand this case.

**Extensibility.**   As mentioned in section 1 and section 2, our proposed idea is a general framework that can be used with *any invertible flow models* (e.g., Residual Flows [23], Flow++ [33]) and *any density model* (e.g., PixelCNN++ [26], FFJORD [22]). Hence, we argue our proposed idea is not limited to specific flow-based model or density model. We expect our framework could show better performance in distribution alignment if it is combined with the aforementioned state-of-the-art components.

## 6   Conclusion

In this paper, we propose a novel variational upper bound on the generalized JSD that leads to a theoretically grounded alignment loss. We then show that this framework unifies previous flow-based distribution alignment approaches and demonstrate the benefits of our approach compared to these prior flow-based methods. In particular, our framework allows a straightforward extension to multi-distribution alignment that could be more parameter efficient than naïvely extending prior approaches. More broadly, we suggest that our AUB metric can be useful as an application-agnostic metric for comparing distribution alignment methods (analogously to how ELBO is used to evaluate density estimation methods). An alignment metric that is not tied to a particular pretrained model (as for FID) or to a particular data type will be critical for systematic progress in unsupervised distribution alignment. We hope this paper provides one step in that direction.

**Acknowledgement.**   The authors thank the anonymous reviewers for their insightful comments and helpful discussion that improved the final paper. The authors acknowledge support from ARL (W911NF-2020-221) and NSF (2212097).

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
