# Supplementary Material

## 1 Proofs

*Proof of Equivalence in Definition 2.1 in the main paper.* While the proof of the equivalence is well-known, we reproduce here for completeness. As a reminder, the KL divergence is defined as:

$$\text{KL}(P, Q) = \mathbb{E}_P[\log \tfrac{P(z)}{Q(z)}] = \mathbb{E}_P[-\log Q(z)] - \mathbb{E}_P[-\log P(z)] = \text{H}_\text{c}(P, Q) - \text{H}(P), \quad (1)$$

where $\text{H}_\text{c}(\cdot, \cdot)$ denotes the cross entropy and $\text{H}(\cdot)$ denotes entropy. Given this, we can now easily derive the equivalence:

$$
\begin{align}
\text{GJSD}_{\boldsymbol{w}}(P_{Z_1}, \cdots, P_{Z_k}) &= \sum_j w_j \, \text{KL}(P_{Z_j}, P_{Z_\text{mix}}) \tag{2}\\
&= \sum_j w_j (\text{H}_\text{c}(P_{Z_j}, P_{Z_\text{mix}}) - \text{H}(P_{Z_j})) \tag{3}\\
&= \sum_j w_j \, \text{H}_\text{c}(P_{Z_j}, P_{Z_\text{mix}}) - \sum_j w_j \, \text{H}(P_{Z_j}) \tag{4}\\
&= \sum_j w_j \mathbb{E}_{P_{Z_j}}[-\log P_{Z_\text{mix}}] - \sum_j w_j \, \text{H}(P_{Z_j}) \tag{5}\\
&= \sum_j w_j \int_{\mathcal{Z}} -P_{Z_j}(z) \log P_{Z_\text{mix}}(z) dz - \sum_j w_j \, \text{H}(P_{Z_j}) \tag{6}\\
&= \int_{\mathcal{Z}} -\sum_j w_j P_{Z_j}(z) \log P_{Z_\text{mix}}(z) dz - \sum_j w_j \, \text{H}(P_{Z_j}) \tag{7}\\
&= \int_{\mathcal{Z}} -P_{Z_\text{mix}}(z) \log P_{Z_\text{mix}}(z) dz - \sum_j w_j \, \text{H}(P_{Z_j}) \tag{8}\\
&= \text{H}(P_{Z_\text{mix}}) - \sum_j w_j \, \text{H}(P_{Z_j}) \,. \tag{9}
\end{align}
$$

$\square$

*Proof of Lemma 2.3 in the main paper.* First, we note the following fact from the standard change of variables formula:

$$
\begin{align}
P_X(\boldsymbol{x}) &= P_Z(T(\boldsymbol{x}))|J_T(\boldsymbol{x})| \\
\Rightarrow P_X(\boldsymbol{x})|J_T(\boldsymbol{x})|^{-1} &= P_Z(T(\boldsymbol{x})) \,. \tag{10}
\end{align}
$$

We can now derive our result using the change of variables for expectations (i.e., LOTUS) and the probability change of variables from above:

$$
\begin{align}
\text{H}(P_Z) &= \mathbb{E}_{P_Z}[-\log P_Z(\boldsymbol{z})] = \mathbb{E}_{P_X}[-\log P_Z(T(\boldsymbol{x}))] \\
&= \mathbb{E}_{P_X}[-\log P_X(\boldsymbol{x})|J_T(\boldsymbol{x})|^{-1}] \\
&= \mathbb{E}_{P_X}[-\log P_X(\boldsymbol{x})] + \mathbb{E}_{P_X}[-\log |J_T(\boldsymbol{x})|^{-1}] \\
&= \text{H}(P_X) + \mathbb{E}_{P_X}[\log |J_T(\boldsymbol{x})|] \,.
\end{align}
$$

$\square$

*Proof of Theorem 2.5 in the main paper.* Given any fixed $Q$, minimizing $\mathcal{L}_\text{AUB}$ decouples into minimizing separate normalizing flow losses where $Q$ is the base distribution. For each normalizing flow, there exists an invertible $T_j$ such that $T_j(X_j) \sim Q$, and this achieves the minimum value of $\mathcal{L}_\text{AUB}$.

More formally,

$$\min_{T_1,\cdots,T_k} \mathcal{L}_{\mathrm{AUB}}(T_1,\cdots,T_k) \tag{11}$$

$$= \min_{T_1,\cdots,T_k} \sum_j w_j \mathbb{E}_{P_{X_j}}[-\log|J_{T_j}(\boldsymbol{x})|\,Q(T_j(\boldsymbol{x}))] \tag{12}$$

$$= \sum_j w_j \min_{T_j} \mathbb{E}_{P_{X_j}}[-\log|J_{T_j}(\boldsymbol{x})|\,Q(T_j(\boldsymbol{x}))] + \mathrm{H}(P_{X_j}) - \mathrm{H}(P_{X_j}) \tag{13}$$

$$= \sum_j w_j \min_{T_j} \mathbb{E}_{P_{X_j}}[-\log|J_{T_j}(\boldsymbol{x})|\,Q(T_j(\boldsymbol{x}))] + \mathrm{H}(P_{X_j}) - \mathbb{E}_{P_{X_j}}[-\log P_{X_j}(\boldsymbol{x})]) \tag{14}$$

$$= \sum_j w_j \mathrm{H}(P_{X_j}) + \sum_j w_j \min_{T_j} \mathbb{E}_{P_{X_j}}[\log \tfrac{P_{X_j}(\boldsymbol{x})|J_{T_j}(\boldsymbol{x})|^{-1}}{Q(T_j(\boldsymbol{x}))}] \tag{15}$$

$$= \sum_j w_j \mathrm{H}(P_{X_j}) + \sum_j w_j \min_{T_j} \mathbb{E}_{P_{X_j}}[\log \tfrac{P_{T_j(X_j)}(T_j(\boldsymbol{x}))}{Q(T_j(\boldsymbol{x}))}] \tag{16}$$

$$= \sum_j w_j \mathrm{H}(P_{X_j}) + \sum_j w_j \min_{T_j} \mathbb{E}_{P_{T_j(X_j)}}[\log \tfrac{P_{T_j(X_j)}(\boldsymbol{z})}{Q(\boldsymbol{z})}] \tag{17}$$

$$= \sum_j w_j \mathrm{H}(P_{X_j}) + \sum_j w_j \min_{T_j} \mathrm{KL}(P_{T_j(X_j)}, Q). \tag{18}$$

Given that $\mathrm{KL}(P,Q) \geq 0$ and equal to 0 if and only if $P = Q$, the global minimum is achieved only if $P_{T_j(X_j)} = Q, \forall j$ and there exist such invertible functions (e.g., the optimal Monge map between $P_{X_j}$ and $Q$ for squared Euclidean transportation cost [1]). Additionally, the optimal value is $\sum_j w_j \mathrm{H}(P_{X_j})$, which is constant with respect to the $T_j$ transformations. $\qquad\square$

## 2 Additional Experiment Details

### 2.1 Toy Dataset Experiment

**LRMF vs. Ours Experiment**

- $T$ for LRMF setup: $T_1$: 8 channel-wise mask for Real-NVP model with $s$ and $t$ derived from 64 hidden channels of fully connected networks. $T_2$: Identity function.
- $T$ for RAUB setup: $T_1$ and $T_1$: 8 channel-wise mask for Real-NVP model with $s$ and $t$ derived from 64 hidden channels of fully connected networks.
- $Q$ for both: A single Gaussian distribution with trainable mean and trainable variances.

**Alignflow vs. Ours Experiment**

- $T$ for both: 2 channel-wise mask for RealNVP model with $s$ and $t$ derived from 8 hidden channels of fully connected networks.
- $Q$ for Alignflow setup: A single fixed normal distribution.
- $Q$ for RAUB setup: A learnable mixture of Gaussian with 3 components.

### 2.2 Tabular Dataset Experiment

Our invertible transformation function $T$ adapts general purpose RealNVP (5) and RealNVP (10) layers with detailed parameters provided in Table 1. Successive coupling layers are concatenated by alternating the format between keeping odd-parity index of the data samples and transforms the even-parity index of the data and vice versa. Note that our density model $Q$ for AUB experiments shares the same architecture of RealNVP (5) throughout the two-domain tabular dataset experiment and RealNVP (10) for multi-domain experiment. For AlignFlow hybrid and adversarial only models, in order to adapt their model for a tabular dataset, we change the invertible model $T\_src$ and $T\_tgt$ to RealNVP(5) and change the discriminator to a fully connected network which has a hidden dimension of 256.

## 3 Multi-domain translation

To illustrate that our method can be easily scaled to more domain distributions even for high dimensional data, we present qualitative examples of translating between every digit and every other

Table 1: RealNVP layers used in tabular experiment section contains $<n\_layers>$ coupling layers. Each coupling layer has two fully connected networks modeling the scaling and shifting function with $<hidden\_dim>$ hidden dimensions and $<n\_hidden>$ hidden layers.

|  | n_layers | hidden_dim | n_hidden | Total number of parameters |
|---|---|---|---|---|
| RealNVP(5) | 5 | 100 | 1 | 1,462,400 |
| RealNVP(10) | 10 | 100 | 2 | 4,540,800 |

Table 2: FID and AUB score for domain alignment task in 10 domains. FID score is calculated by average across all paired translations and AUB score is shown in nats.

|  | FID | AUB |
|---|---|---|
| AlignFlow (MLE) | 49.82 | -4661.05 |
| Ours | 43.25 | -4715.02 |

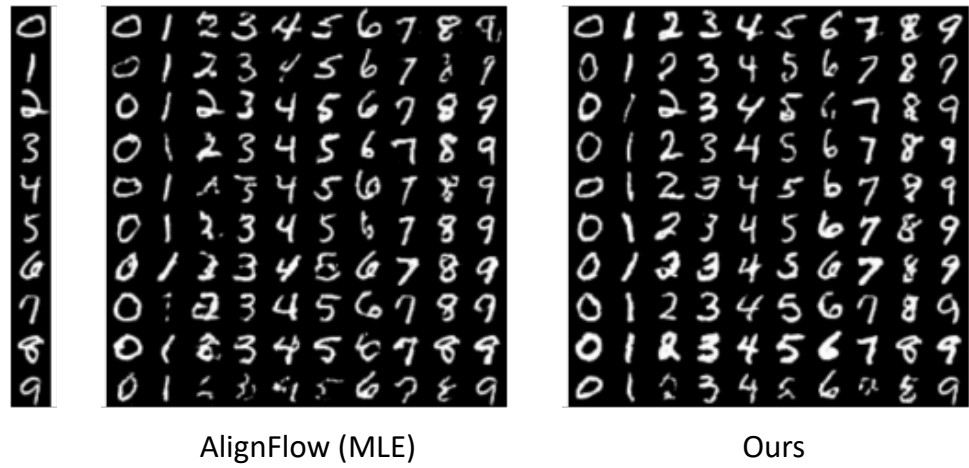

AlignFlow (MLE)                    Ours

Figure 1: Qualitative comparison on translation results across 10 classes with AlignFlow and ours.

digit for MNIST in Fig. 1 with quantitative performances in Table 2. Note that we omit LRMF in this experiment because the multi-domain situation is hard to deal with for LRMF setup due to LRMF's two distribution setup and assymetric model structure.

As shown in Table 2, our approach shows better performance in terms of FID and AUB than AlignFlow since our learnable density estimator can model more complex distribution than fixed simple density model in AlignFlow. In other words, the shared space of AlignFlow is limited because of the fixed simple density model.

The superiority of our method compared to AlignFlow in multiple-domain translation can also be verified through qualitative comparisons in Fig. 1. The leftmost column is an input image and the second and third macro columns are the results from AlignFlow and ours. By forwarding a given $k$-th latent $T_k(x_k)$ into 10 inverse transformation functions, respectively. It is easy to observe that ours has more clear results in most cases than baseline. Moreover, our model shows better performance in maintaining the original identity (e.g., width and type of a stroke) than the baseline, as seen in fifth, eighth and ninth rows. This is because we jointly train our transformation functions with a learnable density model, while AlignFlow independently train their transformation functions. This benefit of our approach may be crucial for other datasets such as human faces [2] where maintaining the original identity is important.

## 4 Implementation on Latent Space

Our model is also capable of generating samples in each domain. One needs to sample from the density model $Q$ to have a latent image $z$ first, and then forward to the inverse function $T_j^{-1}$ to get

the image sampled in $j^{th}$ domain. Examples of generated images for each domain in MNIST data are shown in Fig. 2. The quality of the generative result also reflects the tightness of the bound between the latent space and the latent density model as illustrated in Eqn. 4.

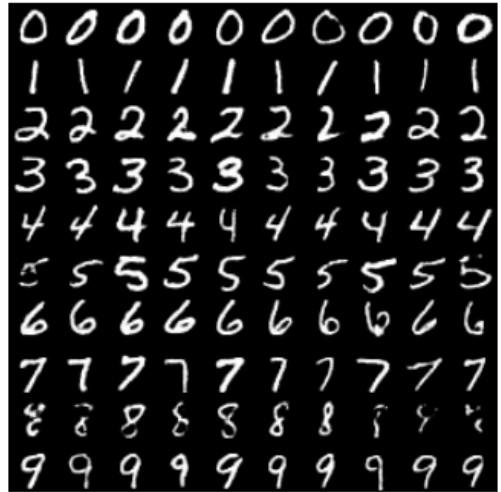

Figure 2: This figure shows the generated images of our model for all domains.

In order to visualize the benefit of our shared latent space, we further perform interpolation in the latent space. We first randomly select two distinct real images in one domain (in this case two '0's), and do a linear interpolation of the selected two images in the latent space. Then we translate all the interpolated images (including the two selected images) to all of the remaining domains to generate 'translated-interpolated' images, i.e., the corresponding interpolations in each of the remaining domains. As shown in Fig. 3, all 'translated-interpolated' results can preserve the trend of the stroke width of the digits from the original interpolated domain. These results suggest that our shared space aligns the domains so that some latent space directions have similar semantic meaning for all domains.

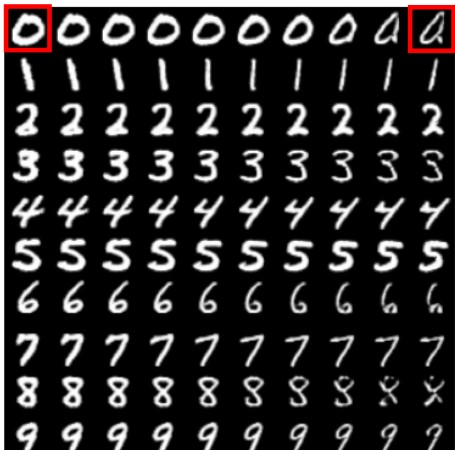

Figure 3: This figure shows the translation results of interpolated images. In the first row, the two images selected by red rectangles are real images from the dataset; and all eight images in between are generated by linear interpolation in the latent space. Starting from the second row, each row contains translated images which are transformed from the same latent vector in the same column and the first row.

# 5 Qualitative results of Domain Adaptation experiments on USPS-MNIST dataset

Here are the translated results that we used in Table 4. The results consistently show that ours show the best performance. Please note that these samples are randomly chosen.

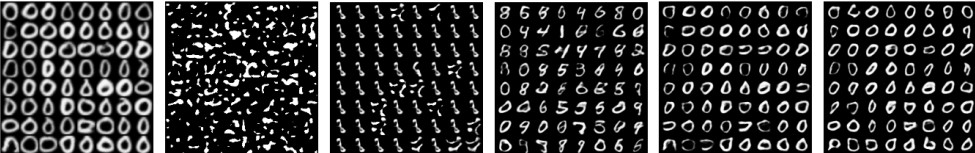

Figure 4: Digit 0 results. From left, USPS data, LRMF, AF (Adv. only), AF (MLE), AF (1e0), AUB (ours)

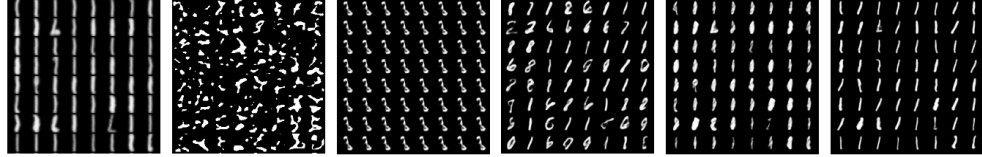

Figure 5: Digit 1 results. From left, USPS data, LRMF, AF (Adv. only), AF (MLE), AF (1e0), AUB (ours)

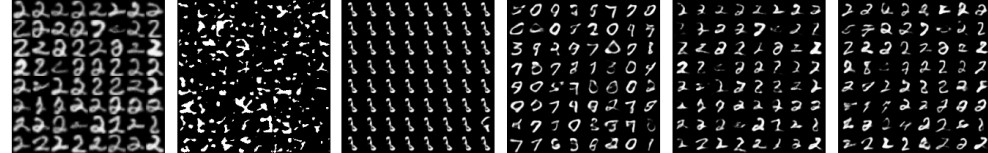

Figure 6: Digit 2 results. From left, USPS data, LRMF, AF (Adv. only), AF (MLE), AF (1e0), AUB (ours)

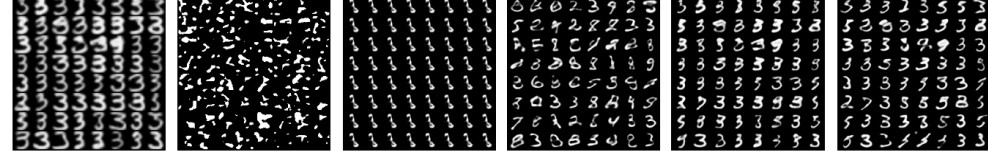

Figure 7: Digit 3 results. From left, USPS data, LRMF, AF (Adv. only), AF (MLE), AF (1e0), AUB (ours)

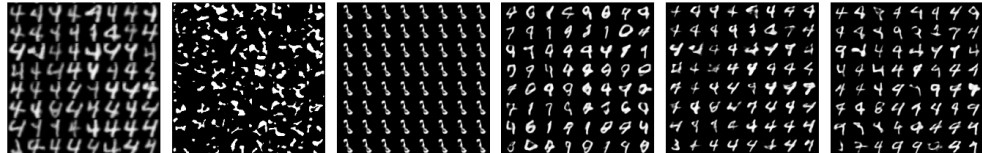

Figure 8: Digit 4 results. From left, USPS data, LRMF, AF (Adv. only), AF (MLE), AF (1e0), AUB (ours)

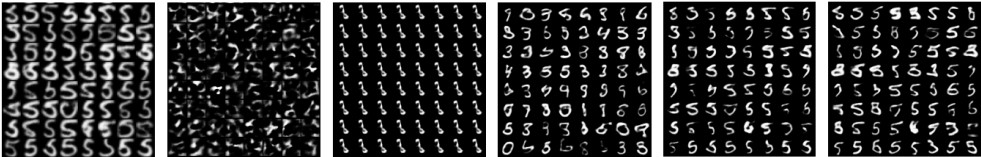

Figure 9: Digit 5 results. From left, USPS data, LRMF, AF (Adv. only), AF (MLE), AF (1e0), AUB (ours)

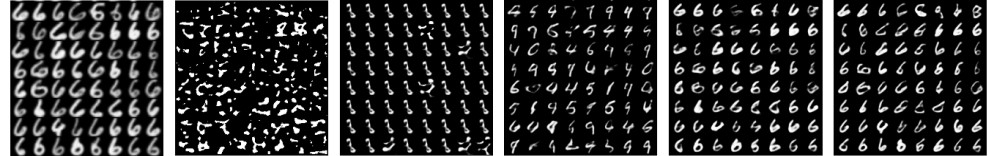

Figure 10: Digit 6 results. From left, USPS data, LRMF, AF (Adv. only), AF (MLE), AF (1e0), AUB (ours)

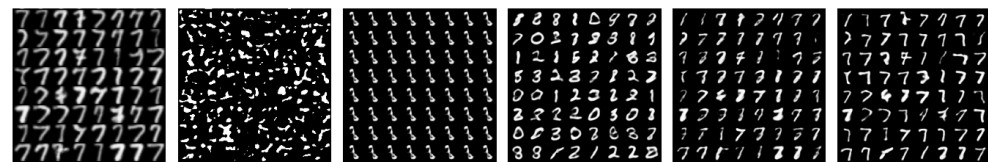

Figure 11: Digit 7 results. From left, USPS data, LRMF, AF (Adv. only), AF (MLE), AF (1e0), AUB (ours)

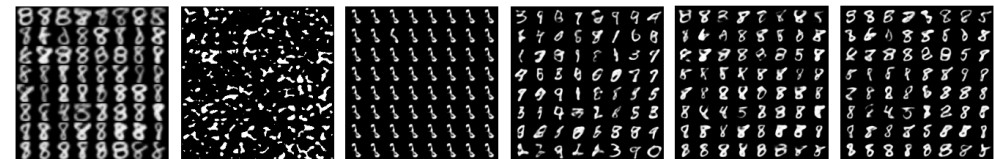

Figure 12: Digit 8 results. From left, USPS data, LRMF, AF (Adv. only), AF (MLE), AF (1e0), AUB (ours)

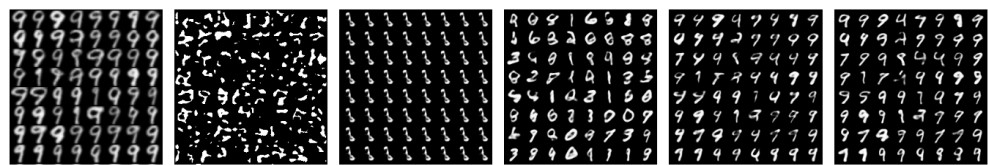

Figure 13: Digit 9 results. From left, USPS data, LRMF, AF (Adv. only), AF (MLE), AF (1e0), AUB (ours)