# OpenReview forum: "Cooperative Distribution Alignment via JSD Upper Bound"
_NeurIPS.cc/2022/Conference — NeurIPS 2022 Accept_

### Official Review · Reviewer_M6As · 2022-07-09

**Rating:** 5
**Confidence:** 3
**Soundness:** 3 good
**Presentation:** 3 good
**Contribution:** 2 fair

**Summary:**

This paper proposes a new objective function to calculate the Generalized Jensen-Shannon Divergence, which can be used to the field of the unsupervised distribution alignment. The experimental results on different tasks show the effectiveness of the proposed objective.

**Questions:**

1. The impact of different values of $w$ on the final performance of the model is not clear.
2. The main part of the article does not refer to Figure 2.
3. As for the objective 6 and Algorithm 1, we can see that the update of $Q$ and $T$ is in a step by step manner, why not update the $Q$ and $T$ in and end-to-end manner,
4. How to guarantee ${P_{{Z_{mix}}}} \in Q$ ？
5. From Figure 5, we can see that the training process of the proposed method in this paper is obviously more stable than that of AlignFlow, I hope the author can give a theoretical analysis.
6. I hope the author can provide the results when using this objective function in the domain adaptation and image generation.

**Limitations:**

The authors have adequately addressed the limitations and potential negative societal impact of their work.

**Strengths And Weaknesses:**

Strengths:
1. The theoretical analysis provided is interesting and make sense.
2. This paper is easy to read.
3. The experimental results are good and adequate.

Weaknesses:
1. The motivation of this paper is vague and unconvincing.
2. The novel of this paper is limited.

---

> ### Author Response · Authors · 2022-08-02
> **Response to Reviewer M6As (Part2)**
>
> **For question 4:** ***How to guarantee $P_{Z_{mix}}\in Q$?***
>
> We first note that, unlike the discussion in the LRMF would suggest, our upper bound holds whether or not $P_{Z_{mix}} \in \mathcal{Q}$.
> Our theory proves that the bound gap is exactly $\min_{Q \in \mathcal{Q}} KL(P_{Z_{mix}}, Q)$ (Theorem 2.2), which is zero if and only if $P_{Z_{mix}} \in \mathcal{Q}$.
> Theoretically, our proofs do rely on the fact that we optimize over the class of *all* invertible functions for $T_j$.  If this is true, then for any $Q$, there exists an invertible maps $T_j$ such that $P_{T_j(X)} = Q$ (for example, the Monge map is invertible and satisfies this property though other invertible maps also exist).
> In practice, we use a parameterized $T_j$ so this condition exact alignment is possible if and only if there exists $T_j$ and $Q\in \mathcal{Q}$ such that $P_{T_j(X)} = Q$ for all $j$, i.e., there only needs to some $T_j$ and one $Q$ such in the model classes that satisfies this constraint.
> More generally, if this does not hold, then the global minimum of AUB may not correspond to perfect alignment.
> The formal proof of Theorem 2.5 is in the appendix.
>
>
> **For question 5:** ***From Figure 5, we can see that the training process of the proposed method in this paper is obviously more stable than that of AlignFlow, I hope the author can give a theoretical analysis.***
>
> Because AlignFlow in Figure 5 is a hybrid loss (i.e., both adversarial and MLE terms), we compare these two AlignFlow terms separately to AUB.  The validation loss in AUB was an ad-hoc combination of these two loss functions and is shown in Figure 5b (see Equation 7. of the AlignFlow paper).
>
> (Adv. loss vs AUB):  The adversarial objective in AlignFlow can have instability in convergence and training as explored in [Lucic et al., 2018; Kurach et al., 2019; Farnia & Ozdaglar, 2020; Nie & Patel, 2020; Wu et al., 2020], while our cooperative training does not have such issues. Please see these references for more theoretic analysis of the instability of GANs.
>
> (AlignFlow(MLE) vs AUB): Theorem 2.5 in our paper mentions "If AUB is minimized over the class of all invertible functions,
> a global minimum of AUB implies that the latent distributions are aligned for all $j \neq j'$. Importantly, this result holds regardless of $Q$."
> Based on this, we expect both AlignFlow (MLE-only) and ours would have stable training and validation losses. However, the difference in ours and AlignFlow (MLE-only) comes from the gap between the true JSD and the approximated joint distribution (c.f., Theorem 2.2 in our paper, formulated as $\min_Q KL(P_{Z_{mix}}, Q))$.
> Because AlignFlow has fixed $Q$, it cannot reduce the gap during the training while ours can reduce the gap with learnable $Q$.
> In summary, we can think both approaches could be stable in training, while ours can show better performance in the alignment because of the tighter gap as empirically demonstrated by our experiments.
>
> [Lucic et al., 2018] Mario Lucic, Karol Kurach, Marcin Michalski, Sylvain Gelly, and Olivier Bousquet. Are GANs Created Equal? A Large-Scale Study. Neurips, 2018
>
> [Kurach et al., 2019] Karol Kurach, Mario Lucic, Xiaohua Zhai, Marcin Michalski, and Sylvain Gelly. The GAN Landscape: Losses, Architectures, Regularization, and Normalization, 2019
>
> [Farnia & Ozdaglar, 2020] Farzan Farnia and Asuman Ozdaglar. Do GANs always have Nash equilibria? In Hal Daumé III and Aarti Singh, editors, ICML, 2020
>
> [Nie & Patel, 2020] Weili Nie and Ankit B Patel. Towards a Better Understanding and Regularization of GAN Training Dynamics. In Uncertainty in Artificial Intelligence (UAI), 2020
>
> [Wu et al., 2020] Yue Wu, Pan Zhou, Andrew G Wilson, Eric Xing, and Zhiting Hu. Improving GAN Training with Probability Ratio Clipping and Sample Reweighting. Neurips, 2018
>
>
> **For question 6:** ***I hope the author can provide the results when using this objective function in the domain adaptation and image generation.***
>
> Thank you for the suggestion. Please refer to the additional experiment in the response to all reviewers.

---

> > ### Author Response · Authors · 2022-08-05
> > **Have we answered your primary concerns?**
> >
> > Thank you for your thoughtful original review.
> > Given our responses, have we adequately addressed your questions and concerns? If so, would you consider updating your review and score? Thank you for your time and consideration.

---

> ### Author Response · Authors · 2022-08-02
> **Response to Reviewer M6As (Part1)**
>
> **For weakness 1:** ***The motivation of this paper is vague and unconvincing.***
>
> Could you explain what you mean or expand on this? The introduction motivates the need for non-adversarial alignment methods both in terms of alternatives to adversarial learning and in terms of validation measures, especially for non-image datasets such as tabular data.
>
> **For weakness 2:** ***The novel of this paper is limited.***
>
> Again, could you explain what you mean or expand on this? We unify prior work via a theoretic upper bound on alignment that shows their strengths and weaknesses (see our extensive discussion in Section 3 and 4.1) and enables novel approaches with multiple $T$ and flexible $Q$. Additionally, because this objective is in fact an upper bound on JSD, it provides a simple and application-agnostic alignment metric for non-image datasets as demonstrated by the tabular experiments. Please also see the response to Reviewers G1Px (and Reviewer tc4i).
>
> **For question 1:** ***The impact of different values of  on the final performance of the model is not clear.***
>
> Thanks for pointing this out. The natural setting of $w$ is to be equal to the prior on domain labels, i.e., if $D$ is the domain label, then this would correspond to setting $w_{j} = P(D=j)$.
> This setting of $w$ makes the GJSD equivalent to the mutual information between the latent representation $z$ and the domain label $D$  (a well-known equivalence).
> However, if all domains are meant to be equally weighted, then a natural setting of $w$ would be $1/k$.
>
> **For question 2:** ***The main part of the article does not refer to Figure 2.***
>
> Thank you. We will be sure to add an appropriate reference.
>
> **For question 3:** ***As for the objective 6 and Algorithm 1, we can see that the update of $Q$ and $T$ is in a step by step manner, why not update the $Q$ and $T$ in and end-to-end manner.:***
>
> We have found empirically that simultaneous optimization could lead to a problematic optimization landscape and local minimum (related to the vanishing gradient issue of GANs).
> In preliminary experiments, we found that while a fixed Gaussian $Q$ (i.e., AlignFlow-MLE) did not perform as well, a completely fitted $Q$ at initialization (e.g., pretrained) also did not perform well because it would get stuck.
> Eventually, we found that alternating minimization initialized at a Gaussian $Q$ seemed to appropriately trade-off these two issues.
> However, we definitely think more exploration is needed in future works.
> For example, motivated by prior works (e.g., [Arjovsky & Bottou, 2017]) that suggest stabilizing GANs by adding noise (particularly when the supports are disjoint) and the success of denoising diffusion models [Ho et al. 2020], we expect that adding noise to the observed and/or latent space could significantly reduce this local minimum or vanishing gradient problem.
> In fact, using the derivation of Theorem 3.3 from [Arjovsky & Bottou, 2017], we could trivially extend the upper bound of Wasserstein-1 distance in Theorem 3.3 by adding Gausian noise.
> Orthogonal to GAN literature, future work could also consider how to explicitly regularize $Q$ such that it would enable appropriate learning.
> For example, weighting the $\log Q(z)$ term by $\lambda$ would effectively be using an (unnormalized) model $\tilde{Q}(z) \propto Q(z)^{\lambda}$.
> Thus, our approach opens up the door for future optimization innovations.
>
> Furthermore, while it is relatively easy to apply simultaneous optimization for a small number of domains, if there are many domains, merging $Q$ and $T$ updates together could yield a memory issue because we need to obtain the joint representation across all the domains (i.e., $z_{mix}$) to update $Q$.
>
> [Arjovsky & Bottou, 2017] Arjovsky, M., & Bottou, L. (2017). Towards Principled Methods for Training Generative Adversarial Networks. In International Conference on Learning Representations (ICLR).
>
> [Ho et al. 2020] Ho, J., Jain, A., & Abbeel, P. (2020). Denoising diffusion probabilistic models. Advances in Neural Information Processing Systems, 33, 6840-6851.

---

### Official Review · Reviewer_G1Px · 2022-07-11

**Rating:** 7
**Confidence:** 4
**Soundness:** 3 good
**Presentation:** 4 excellent
**Contribution:** 3 good

**Summary:**

This paper addresses the problem of alignment of multiple distributions. The authors propose a new method for distribution alignment based on invertible flow models and a new training objective which gives an upper bound on the generalized Jensen-Shannon divergence (GJSD) between the distributions. The method learns invertible transformations (one per each distribution) from the data space to a shared latent space. The invertible mappings are optimized so that the induced latent space distributions are as close as possible to a variational distribution Q which is chosen from an appropriate family of distributions. The parameters of the variational distribution Q are optimized (via maximum likelihood) so that Q approaches the mixture of the latent space distributions. An important feature of the approach is that the invertible transformations and the variational distribution are trained cooperatively meaning that the training is cast as  a min-min game in contrast to min-max games arising in adversarial alignment approaches.

The authors derive a variational upper bound on the GJSD and convert this upper bound to a loss function which can be computed (and optimized) using only samples from the original distributions. The authors prove that global minima of the objective correspond to invertible mappings that align the distributions. The authors also explain that previous flow based methods (AlignFlow with MLE objective only and LRMF) can be seen as special cases of the proposed approach. Finally, the authors suggest that the proposed training objective can be used as a relative metric for evaluation of alignment methods (similar to how ELBO is used as a metric for density estimation).

The authors conduct experiments on toy data, UCI tabular datasets, and MNIST image dataset. The proposed approach is shown to outperform the AlignFlow and LRMF in terms of alignment, parameter efficiency, translation quality, and training stability.


**Questions:**

Q1: I believe that Theorem 2.5 can be formulated as “global minima of L_AUB is attained if and only if the distributions are aligned”. So, the statement of the theorem can be stronger (implication both ways rather than only “global minima implies alignment”). The current proof already confirms the implication in the opposite direction.

Q2: (line 125) “|J_T(x)| is the determinant of the Jacobian of T” -> “|J_T(x)| is the absolute value of the determinant of the Jacobian of T”

Q3: legends in Figure 3 are barely readable. What do the green ellipses represent?

Q4: (line 227) what does “identity-initialized” Q mean?

Q5: The role of the variational distribution Q is still not completely clear to me after reading the paper. In principle the alignment of P_1, …, P_k can be achieved by optimizing the AUB loss with any fixed Q (as AUB is equivalent to the sum of KL(P_j, Q)). I understand that when Q minimizes the AUB loss (cooperatively with P_1, … P_k) it is supposed to help optimization over P_1, …, P_k. The optimal Q fits the mixture of P_1, … P_k exactly, and when Q reaches that optimum, the AUB becomes exactly equal to the GJSD between the distributions. Given this interaction between P_1, …, P_k and Q, I have two questions:

Q5.1: Is it really important that AUB loss gives an upper bound on GJSD? It seems that this fact does not affect the ability of the method based on AUB minimization to align the distributions. I understand that having a limited family of variational distributions Q might be problematic (as the results of AlignFlow on synthetic data suggest). However, it is still not clear what is the best choice of Q (since any Q should work in principle).

Q5.2: It seems that optimization of L_AUB w.r.t. Q could lead to a problematic optimization landscape for P_1, … P_k. Assuming that Q reaches its goal and fits the mixture of P_1, …, P_k, such Q most likely will be multimodal (especially if distributions P_i have disjoint supports at the initialization) with each mode corresponding to one of P_i. Then, the optimization objective w.r.t. P_i is equivalent to reverse-KL (KL(P_i, Q) and we are optimizing over P_i). The reverse-KL optimization is known to have a bias towards locally fitting modes of the target distribution, so P_i being a mode of the mixture might be stuck in a local optima when Q fits the mixture.


**Limitations:**

I do not foresee any direct negative societal impact of this work.

**Strengths And Weaknesses:**

S: strength W: weakness

S1: The paper contributes to the line of work on unsupervised distribution alignment via flow models and proposes a novel (to the best of my knowledge) approach which generalizes some of the existing techniques.

S2: The paper is technically sound. The theoretical results and the proofs are correct. The chosen experimental setups and metrics are adequate.

S3: The paper is clearly-written and well-organized. The presentation of the theoretical motivation behind the method is precise. The experiments are described in detail and the results are reported clearly.

W1: Paper is missing references to a related line of work (references [1] and [2] below) on non-adversarial alignment methods via iterative flow models based on sliced optimal transport.

[1] Dai, Biwei, and Uros Seljak. "Sliced Iterative Normalizing Flows." International Conference on Machine Learning, 2021

[2] Zhou, Zeyu, et al. "Iterative Alignment Flows." International Conference on Artificial Intelligence and Statistics, 2022.

---

> ### Author Response · Authors · 2022-08-02
> **Response to Reviewer G1Px (Part2)**
>
> **For question 5:** ***The role of the variational distribution Q ... I have two questions:***
>
> Thank you for your comments here. You have a excellent understanding of our paper.
>
> **For question 5.1:** ***Is it really important that ... best choice of Q (since any Q should work in principle).***
>
> This is an insightful question. To clarify, we are assuming that your first question about "Is it really important..." means something like "Why not perform MLE for each domain with a shared latent distribution?"  This could lead to alignment without knowing the connection to JSD. Please correct us if we have misunderstood your question here. We suggest a few reasons why it is useful.
>
> First, the bound theoretically grounds the alignment approach as it is not obvious from looking at the AUB objective.
> Second, the bound clarifies the role of the shared $Q$ distribution as tightening the upper bound; this may helpful for developing "regularized" alignment objectives where the bound is intentionally made looser (e.g., by adding noise in the latent space).
> Third and perhaps most importantly, our bound paves the way for future alignment upper bounds, similar to how the original GAN paper (which showed a connection between JSD and standard cross entropy classifiers) was generalized from JSD to Wasserstein-1 [Arjovsky et al., 2017] and $f$ divergences [Nowozin et al., 2016].
>
> [Arjovsky et al., 2017] Arjovsky, M., Chintala, S., & Bottou, L. (2017, July). Wasserstein generative adversarial networks. In International conference on machine learning (pp. 214-223). PMLR.
>
> [Nowozin et al., 2016] Nowozin, S., Cseke, B., & Tomioka, R. (2016). f-gan: Training generative neural samplers using variational divergence minimization. Advances in neural information processing systems, 29.
>
> Regarding the choice of $Q$, we show empirically that a flexible $Q$ is better than a fixed Gaussian $Q$ (i.e., AlignFlow-MLE).
> More generally, we expect the choice of $Q$ to be application and context-specific.
> We believe that one possible good choice of the density model $Q$ is one that is powerful enough to model the general structure (e.g., image features) across different domains so that the transformation function $T$ only needs to model the information distinct to the current domain (e.g., rotations, background).
> In this context, our method could leverage $Q$ models developed specifically for images.
> Or, if alignment is being used as a constraint in another task (e.g., fair representation learning [Balunovic et al., 2021]), then a flexible $Q$ is necessary to enforce that the latent representations are aligned **and** satisfy additional application-specific properties (e.g., simple classification on top of the latent representation).
>
> [Balunovic et al., 2021] Balunovic, M., Ruoss, A., & Vechev, M. (2021, September). Fair normalizing flows. In International Conference on Learning Representations (ICLR).
>
>
> **For question 5.2:** ***It seems that optimization of ... stuck in a local optima when Q fits the mixture.***
>
> This is another excellent observation!  Yes, indeed we have found empirically that this could lead to a problematic optimization landscape and local minimum (related to the vanishing gradient issue of GANs).
> In preliminary experiments, we found that while a fixed Gaussian $Q$ (i.e., AlignFlow-MLE) did not perform as well, a completely fitted $Q$ at initialization (e.g., pretrained) also did not perform well because it would get stuck.
> Eventually, we found that alternating minimization initialized at a Gaussian $Q$ seemed to appropriately trade-off these two issues.
> However, we definitely think more exploration is needed in future works.
> For example, motivated by prior works (e.g., [Arjovsky & Bottou, 2017]) that suggest stabilizing GANs by adding noise (particularly when the supports are disjoint) and the success of denoising diffusion models [Ho et al. 2020], we expect that adding noise to the observed and/or latent space could significantly reduce this local minimum or vanishing gradient problem.
> In fact, using the derivation of Theorem 3.3 from [Arjovsky & Bottou, 2017], we could trivially extend the upper bound of Wasserstein-1 distance in Theorem 3.3 by adding Gausian noise.
> Orthogonal to GAN literature, future work could also consider how to explicitly regularize $Q$ such that it would enable appropriate learning.
> For example, weighting the $\log Q(z)$ term by $\lambda$ would effectively be using an (unnormalized) model $\tilde{Q}(z) \propto Q(z)^{\lambda}$.
> Thus, our approach opens up the door for future optimization innovations.
>
> [Arjovsky & Bottou, 2017] Arjovsky, M., & Bottou, L. (2017). Towards Principled Methods for Training Generative Adversarial Networks. In International Conference on Learning Representations (ICLR).
>
> [Ho et al. 2020] Ho, J., Jain, A., & Abbeel, P. (2020). Denoising diffusion probabilistic models. Advances in Neural Information Processing Systems, 33, 6840-6851.

---

> > ### Comment · Reviewer_G1Px · 2022-08-05
> > **Author Rebuttal Acknowledgement and further comments regarding GJSD upper bound**
> >
> > Thank you for thoroughly addressing my questions. Your response helped me better understand the results of your paper.
> >
> > I would like to make a comment regarding Q 5.1.
> >
> > I do not completely agree with you claim that "the bound theoretically grounds the alignment approach as it is not obvious from looking at the AUB objective." Equation (18) in supplementary material states that that the AUB objective is equivalent to the sum of KL divergences KL(P_{T_j(X_j)}, Q) up to a constant which does not depend on the transformations. This observation is enough to ground the alignment approach and it does not require reasoning about the GJSD upper bound. This fact is what initially made me confused about the importance of the GJSD upper bound argument.
> >
> > Perhaps a better way to formulate Q 5.1 is "Why should one focus on GJSD (and it's upper bound) specifically as measure of alignment?".
> > There are many different ways to metrics for measuring degree of misalignment of the distributions. The sum of KL divergences in eq (18) is a valid measure of alignment for any Q (in the sense that it is zero iff the distributions are aligned). What I was trying to understand is what makes GJSD a more important/interesting divergence.
> >
> > Now I realize that there is a significant difference between GJSD and sum of KL divergences in eq(18). GJSD compares distributions P_1, ... P_k between themselves. It is equivalent to asking how is each P_i different from the mixture of all P_1 .. P_k. The sum of KL divergences in eq (18) compares each of distributions P_1 ... P_k to another auxiliary distribution Q. While the minimization of both objectives leads to alignment of distributions one would argue that GJSD is a more meaningful measure as it compares P_1 ... P_k with each other and does not involve comparison with another external distribution Q. One could imagine a situation where P_1 = ... = P_k = P and GJSD is zero but if the variational distribution Q \neq P then the sum of KL divergences (eq 18) is not zero. If we were to minimize AUB w.r.t. P_1 ... P_k starting from such configuration, we would destroy the alignment of P_1 ... P_k and try make them all aligned with Q. In this situation it is clearly helpful to optimize AUB w.r.t. Q as well to make the GJSD upper bound tight and preserve the aligned configuration at the initialization.
> >
> > Please feel free correct me if you see any problems with my arguments above.
> >
> > In summary, the reason for Q 5.1 was my confusion about the role of GJSD upper bound after reading the paper. The misunderstanding is resolved now. I would encourage the authors to incorporate ideas from our discussion in the next revision of the paper in order to help future readers better understand the role of Q and importance of the fact that AUB objective is an upper bound on the GJSD. I believe this topic needs some clarification as even the authors themselves are a little bit confused in their response (as I mentioned above the comment about the bound "grounding the alignment approach" is not entirely on point).

---

> > > ### Author Response · Authors · 2022-08-05
> > > **Response to Reviewer G1Px**
> > >
> > > Thank you for sharing us the insightful idea.
> > > We will thoroughly review and revise the part we have discussed about for preventing any possible misunderstandings on our paper.
> > > Again, thank you for your time and considerations to make our paper better.

---

> ### Author Response · Authors · 2022-08-02
> **Response to Reviewer G1Px (Part1)**
>
> **For weakness 1:** ***Paper is missing references to a related line of work (references [1] and [2] below) on non-adversarial alignment methods via iterative flow models based on sliced optimal transport.***
>
> Thank you so much for these related works!  We will definitely add these to our final paper.  These are indeed alternative approaches to distribution alignment via iteratively building up a deep model via simpler maps.
> While they do not use standard adversarial learning as in GANs, their problem formulation can still be seen as adversarial in nature because they are minimizing the max sliced Wasserstein distance, which is itself a maximization problem; thus the resulting problem is actually a min over maps and a max over sliced Wasserstein directions.
> This adversarial connection is made more clear in [Zhou et al. 2022] but applies to [Dai & Seljak, 2021] as well.
> The key difference with our work is that, for the task of distribution alignment, neither method can prove that their algorithm reduces a global divergence measure---though they do reduce the ``local'' divergence at each iteration.
> Indeed, it would be interesting future work to see if our upper bound could be used to prove that these iterative methods are actually reducing an upper bound and the upper bound gets tighter at each iteration.
> Perhaps our theory can enable better theory for these methods, which behave reasonably well in practice despite their simplicity.
>
> **For question 1&2:** ***1) I believe that Theorem 2.5 can be formulated ... confirms the implication in the opposite direction. 2) (line 125) “$|J_T(x)|$ is the determinant of the Jacobian of $T$ -> $|J_T(x)|$ is the absolute value of the determinant of the Jacobian of $T$***
>
> Thanks for pointing this out! Yes, we can definitely strengthen the theorem statement as you mention. And yes, thank you for noticing the missing ``absolute value'' in the Jacobian.  We will correct in the final version.
>
> **For question 3:** ***legends in Figure 3 are barely readable. What do the green ellipses represent?***
>
> Since we use mixture of Gaussian (MoG) model as the density distribution, the green ellipses represent the contour of each Gaussian component in one standard deviation in the MoG models.
>
> **For question 4:** ***(line 227) what does “identity-initialized” Q mean?***
>
> We apologize for the lack of clarity. “identity-initialized $Q$” means that the flow model has its transformation function $T$ initialized as identity function (i.e., *at initialization*, the $Q$ distribution is a Gaussian in our experiments).
> It would be more precise to say that $Q$ has a flow transformation $T$ that is initialized at the identity function.

---

### Official Review · Reviewer_tc4i · 2022-07-12

**Rating:** 6
**Confidence:** 4
**Soundness:** 4 excellent
**Presentation:** 3 good
**Contribution:** 3 good

**Summary:**

Authors propose a novel non-adversarial Alignment Upper Bound (AUB) objective that uses normalizing flows to learn invertible mappings from respective domains to a shared latent space modeled explicitly by another parametric density model Q. The objective, up to a constant, equals the weighted log-likelihood of multiple flow densities each fitted to samples from the respective domain using a shared prior Q that is also optimized. Authors show that the proposed objective upper bounds the JSD and leads to a minimization alignment problem. Authors show that the objective is a generalization of LRMF and AlignFlow, and illustrate specific toy configurations in which equivalent LRMF and AlignFlow models fail. Authors show that their method achieves better alignment on tabular data, and achieves better image realism when transforming across MNIST digit classes compared to both LRMF and AlignFlow.

**Questions:**

See concerns above.

**Limitations:**

Authors discuss that the performance of their approach is limited by the quality of the underlying density model.

**Strengths And Weaknesses:**

1. The main idea is well-motivated and is presented very well.
2. The relation to prior work is established.
3. The derivations are sound.
4. Toy experiments highlight specific cases in which prior work fails.

Nevertheless, I still have some concerns:

1. L168 “AlignFlow without adversarial loss terms is a special case of our method for two distributions where the density model class Q only contains the standard normal distribution (i.e., a singleton class).”, In the “Maximum Likelihood Estimation” section of the AlignFlow paper, Grover et al. discuss that parameter sharing across early layers might be helpful ("Instead, we can share parameters between the two mappings"). In my understanding, an MLE-based AlignFlow with shared layers would be equivalent to an AUB with flow-based Q - is that correct?

2. L178 “[LRMF does] not uncover the connection of the objective as an upper bound on JSD”. Looking at the LRMF paper, the sub-section “Relation to Jensen-Shannon divergence and GANs” seems to establish some relationship between the introduced objective the JSD?

3. Both tabular and digit experiments tell very little about the semantic quality of the alignment. The MNIST digit class transformation task is somewhat contrived. Is there a chance you could include at least an experiment with USPS-to-MNIST adaptation with post-adaptation classification scores?

Some minor comments:

The equation introduced in L119-120 is critical for going from (3) to (4), but can be easily missed. I would have appreciated it if the eq in L119-120 had its own number and was referenced around L130.

There seems to be a pair of parentheses missing in eq (4-5) - the log() should be applied to the product of J and Q, but log Q |J| reads as if log Q is weighed by |J|.

---

> ### Author Response · Authors · 2022-08-02
> **Response to Reviewer tc4i**
>
> **For concern 1:** ***L168 “AlignFlow without adversarial loss terms is a special case ... would be equivalent to an AUB with flow-based Q - is that correct?"***
>
> Yes, that is correct. By using the same architecture of $T$ and $Q$, our model can be treated as MLE-based AlignFlow model in which the transformation functions share the *last* few layers (in AlignFlow, it is unclear if the sharing would be the first few or last few layers). However, we note that our approach allows for $Q$ that are not flows, e.g.,  autoregressive densities or mixture models as in our toy experiments that even have alternative learning algorithms than SGD.
>
> **For concern 2:** ***L178 “[LRMF does] not uncover the connection of the objective as an upper bound on JSD”. Looking at the LRMF paper, the sub-section “Relation to Jensen-Shannon divergence and GANs” seems to establish some relationship between the introduced objective the JSD?***
>
> Indeed, LRMF did discuss a connection with JSD but only as "biased estimates of JSD" (p. 5 of LRMF NeurIPS paper), rather than a theoretic *upper bound* of JSD.
> Additionally, our bound holds *regardless of the model class $Q$*, whereas the LRMF-JSD discussion further assumes that the model class is ``convex'', i.e., that for any two densities in the class, their mixture is also in the class---an unusual and possibly unrealistic assumption. This assumption is used to prove a relationship to a closed-form solution to the inner problem of the adversarial JSD form with a "regularized discriminator".
> Again, our theory does not make any assumptions on the model class of $Q$ and yet still proves that the objective is a true upper bound on JSD.
> Ultimately, our paper is a complementary discussion to LRMF that generalizes and clarifies its objective function.
>
> **For concern 3:** ***Both tabular and digit experiments tell very little about the semantic quality of the alignment. The MNIST digit class transformation task is somewhat contrived. Is there a chance you could include at least an experiment with USPS-to-MNIST adaptation with post-adaptation classification scores?***
>
> Thank you for the recommendation to improve out paper. Please refer to the additional experiment.
>
> **For two minor comments:** ***1) The equation introduced in L119-120 is critical for going from (3) to (4), but can be easily missed. I would have appreciated it if the eq in L119-120 had its own number and was referenced around L130. 2)There seems to be a pair of parentheses missing in eq (4-5) - the log() should be applied to the product of J and Q, but log Q |J| reads as if log Q is weighed by |J|.***
>
> We will thoroughly revise our paper following the reviewer's advice.

---

> > ### Author Response · Authors · 2022-08-05
> > **Have we answered your primary concerns?**
> >
> > Thank you for your thoughtful original review.
> > Given our responses, have we adequately addressed your questions and concerns? If so, would you consider updating your review and score? Thank you for your time and consideration.

---

> > ### Comment · Reviewer_tc4i · 2022-08-07
> > **response**
> >
> > I am satisfied with responses that authors have provided. When authors upload a revision of their manuscript with updated related work section that reflects comments above, I am happy to increase my rating to accept.
> >
> > As an afterthought, one more point that I suggest authors to discuss explicitly in their manuscript: it seems that LRMF authors ague that the relationship to JSD is what causes the vanishing of generator gradients both in LMRF and GANs since JSD is locally constant if distributions are spread far apart. Given that AUB is even more strongly connected to JSD, how come it seems to not exhibit any vanishing of generator gradients?

---

> > > ### Author Response · Authors · 2022-08-09
> > > **Revision of manuscript uploaded**
> > >
> > > We just finish revising the manuscript with more detailed arguments on the "Relationship to Prior Works" section from our discussions above along with suggestions from other reviewers. We also include our additional DA experiment in the supplementary material.
> > >
> > > In addition, we add the vanishing gradient discussion inside the "Discussion" section: *"Alternating minimization against vanishing gradient problem: the AUB can suffer from a vanishing gradient if the distributions are disjoint and $Q$ is fit very well. However, any $Q$ that does not match $P_{Z_{mix}}$ provides an upper bound that we conjecture is smoother than the true JSD. Thus, in experiments, we have found that alternating minimization does not suffer from vanishing gradients because $Q$ is not fully fitted at each step but provides a smooth upper bound. Yet, deeper theoretic and empirical analysis is needed in future work to fully understand this case."*
> > >
> > > Thanks again for sharing insightful ideas with us!

---

### Official Review · Reviewer_fkCG · 2022-07-12

**Rating:** 4
**Confidence:** 4
**Soundness:** 2 fair
**Presentation:** 3 good
**Contribution:** 2 fair

**Summary:**

This manuscript proposed a framework for flow-based unsupervised dataset/distribution alignment, where a variational objective was derived and is shown to upper bound the JSD. Based on the analysis, the authors derive results linking deep techniques for UDA and motivated their approach. Unlike the adversarial training-based UDA methods, this work adopted the flows-based methods to minimize the JSD upper bound and has been shown tighter than other methods. The paper is generally well written and easy to follow the main idea.

**Questions:**

1. The proposed methodology has cooperative training (min-min optimization). Will this optimization be easy to solve, considering that the method requires the transformations should be invertible?

2. Since this work claims to opt for the cooperative objective rather than adversarial training ones, the benefits of this choice should be well-explained.

3. I would expect to see more comparisons with recent strong baselines in DA (especially those with adversarial training objectives) with more challenging datasets (e.g. Office-31, Office-Home or DomainNet etc.)

Minor questions:
What is the exact meaning of $Z_{mix}$ in Theorem 2.2?

**Limitations:**

The current evaluations are only conducted for some simple datasets, making it hard to check the empirical effectiveness.

**Strengths And Weaknesses:**

Strengths:

1. This paper investigates an interesting flow-based distribution alignment method. It provides good insights into the flow-based approaches, which is a promising research direction.

2. The proposed work is well-motivated and the analysis seems to be technically sound.

Weakness:
1. The cooperative training, which looks like a min-min optimization, seems to lead the Q model to be difficult to train since the proposed methods require the transformations must be invertible. In this respect, I think adversarial training might be better. I would expect the authors to provide further explanations/ discussions for this point.

2. The empirical evaluations are not sufficient, I would expect to see more comparisons with recent strong baselines in DA (especially those with adversarial training objectives) with more challenging datasets (e.g. Office-31, Office-Home or DomainNet etc.)

---

> ### Author Response · Authors · 2022-08-02
> **Response to Reviewer fkCG**
>
> **For questions 1&2: 1)** ***The proposed methodology has cooperative training (min-min optimization). Will this optimization be easy to solve, considering that the method requires the transformations should be invertible? Since this work claims to opt for the cooperative objective rather than adversarial training ones, the benefits of this choice should be well-explained. 2) Since this work claims to opt for the cooperative objective rather than adversarial training ones, the benefits of this choice should be well-explained.***
>
> We agree that the training our proposed method is not easy because of 1) the limited choice for the model because of invertibility and 2) the additionally introduced density model. However, as shown in previous research papers [Lucic et al., 2018; Kurach et al., 2019; Farnia & Ozdaglar, 2020; Nie & Patel, 2020; Wu et al., 2020], adversarial training can be hard in the perspective of training stability and lack of the domain-agnostic metric (e.g., we cannot know when to stop the training especially for tabular datasets). Our method can measure validation performance by computing exact likelihood because of the invertibility. Moreover, it is widely known that mode collapse can happen in adversarial training unless careful hyperparameter tuning is done. As discussed in our paper, we argue that both adversarial methods and cooperative methods have their pros and cons. Our proposed idea is a fundamentally different.
>
> [Lucic et al., 2018] Mario Lucic, Karol Kurach, Marcin Michalski, Sylvain Gelly, and Olivier Bousquet. Are GANs Created Equal? A Large-Scale Study. Neurips, 2018
>
> [Kurach et al., 2019] Karol Kurach, Mario Lucic, Xiaohua Zhai, Marcin Michalski, and Sylvain Gelly. The GAN Landscape: Losses, Architectures, Regularization, and Normalization, 2019
>
> [Farnia & Ozdaglar, 2020] Farzan Farnia and Asuman Ozdaglar. Do GANs always have Nash equilibria? In Hal Daumé III and Aarti Singh, editors, ICML, 2020
>
> [Nie & Patel, 2020] Weili Nie and Ankit B Patel. Towards a Better Understanding and Regularization of GAN Training Dynamics. In Uncertainty in Artificial Intelligence (UAI), 2020
>
> [Wu et al., 2020] Yue Wu, Pan Zhou, Andrew G Wilson, Eric Xing, and Zhiting Hu. Improving GAN Training with Probability Ratio Clipping and Sample Reweighting. Neurips, 2018
>
>
> **For question 3:** ***I would expect to see more comparisons with recent strong baselines in DA (especially those with adversarial training objectives) with more challenging datasets (e.g. Office-31, Office-Home or DomainNet etc.)***
>
>    1. We provide the additional DA experiment above in the comment to all reviewers. Our main contributions would be 1) proposing a theoretically grounded fundamental method and 2) validating its performance in a relatively simple but diverse domain dataset. Achieving State-of-the-art performance or dealing with challenging dataset is not our primary goal and is out of our scope.
>    2. Please understand that the main contribution of this paper is by introducing an application-agnostic measure that can intrinsically evaluate domain alignment. Such measures can be further used as a loss function for training distribution alignment. Domain Adaptation (DA) tasks, on the other hand, pay more attention to preserving the label information while translating between domains. Therefore, our model is not designed for DA experiments, but it is definitely an interesting future research direction.
>
> **For minor questions:** ***What is the exact meaning of $Z_{mix}$ in Theorem 2.2?***
>
> $P_{Z_{mix}}$ is defined as weighted sum of domain-specific $P_Z$, i.e., a mixture distribution, as explained in line 72.

---

> > ### Author Response · Authors · 2022-08-05
> > **Have we answered your primary concerns?**
> >
> > Thank you for your thoughtful original review.
> > Given our responses, have we adequately addressed your questions and concerns? If so, would you consider updating your review and score? Thank you for your time and consideration.

---

> > ### Comment · Reviewer_fkCG · 2022-08-08
> > **Thanks for the responses**
> >
> > I've read the rebuttal and appreciate the authors' efforts to clarify the concerns. I agree with the authors' answers to my questions about min-min optimization. However, I think the additional experiments on simple digits experiments are not sufficient. Thus, I would keep my original rating, and I encourage the authors to also focus more on the empirical validations.

---

### Author Response · Authors · 2022-08-02
**Response to all Reviewers**

## Additional Experiment results:
We additionally conducted MNIST to USPS Domain Adaptation experiments for verifying our alignment performance compared to the baseline models.
Due to the limited rebuttal time, we performed domain alignment on a latent representation of the dataset obtained from a pretrained Variational Autoencoder model (VAE) trained jointly on both MNIST and USPS images (no labels were used to train this VAE). Classification score is therefore measured in the latent space as well.
(We set the latent dimension to have size 32 as opposed to 786(MNIST) and 256(USPS) of the original datasets.)

Specifically, we first pretrain a VAE model and extract the latent representations of training and validation images by forwarding them through the pretrained encoder.
The rest of the experiment is conducted on these latent representations.
Following the typical DA evaluation protocol, we train the classifier with source domain data and evaluate the performance by translating the target domain data onto the source domain.
Image translation results can be obtained by forwarding the latent translated results into the pretrained decoder. Please refer to [this link](https://anonymous.4open.science/r/Neurips-22-review-2C44/README.md) for qualitative comparisons.

|          | LRMF  | AF(Adv.only)  | AF(1e-2)  | AF(1e-1)  | AF(1e0)  | AF(1e1) | AF(MLE) | AUB(ours) |
|----------|-------|:-------------:|:---------:|:---------:|:--------:|:-------:|:-------:|:---------:|
| Accuracy | 12.7% |     11.1%     |   16.6%   |   63.8%   |  68.6%   |  35.7%  |   22%   | **77.5%** |

Table1: Test classification accuracies for domain adaptation from USPS to MNIST. The upperbound (target only) is 98.0%.
AF (number) means the coefficient of MLE term in their hybrid objective. Adversarial methods (i.e AlignFlow Hybrid/AlignFlow Adv. only) are set to stop at 200 epochs.

The rightmost column of the table is our result, and other columns show the baseline performances.
Overall from both quantitative and qualitative results, ours shows the best results among all the baselines. We believe this result is further empirical evidence that our novel cooperative training can be comparable to the adversarial training.
As we argued in the paper, adversarial loss shows mode collapse, e.g., AF(Adv. only), AF(1e-2). without careful hyperparameter tuning.
The other thing to note is AF (MLE) works for domain alignment itself (i.e., the translated result is in MNIST domain), but it does not maintain the class information. We think the class maintaining issue comes from their simple and fixed $Q$ distribution where arbitrary rotations of the latent space have equivalent likelihoods throughout the whole training process. Class information is lost while transforming from/to such $Q$ distribution. On the other hand, our learnable $Q$ guides the alignment training process such that class information is partially preserved during transformation.
LRMF failed to align since the $Q$ distribution is not complex enough to model the same density class of the target distribution.
This is the consistent result with our toy dataset experiment (Fig3, (a)-(c)). LRMF does not work well with the simple $Q$ while ours can align two different distributions with the simple $Q$ because of the shared space.
Please note that all models in the table use relatively the same number of model parameters.

---

### Comment · Area_Chair_W4qF · 2022-08-06
**Please reply to rebuttals**

Dear Reviewers,

Thanks for your work reviewing this paper. There are only a few days left for discussing with the authors.

Please read the authors' rebuttals and **explain why you decided to keep your score as is or why you updated it.** (not just click on the "acknowledge" button).  It is very frustrating for authors to be completely ignored.

Hence, we urge you to read and reply ASAP.

Thanks again,

AC

---

### Meta-Review · Area_Chair_W4qF · 2022-08-26

**Recommendation:** Accept
**Confidence:** Certain

**Metareview:**

This paper addresses the problem of the alignment of multiple distributions. The authors propose a new method for distribution alignment based on invertible flow models and a new training objective that gives an upper bound on the generalized Jensen-Shannon divergence (GJSD) between the distributions.   The authors derive a variational upper bound on the GJSD and convert this upper bound to a loss function.  They prove that global minima of the objective correspond to invertible mappings that align the distributions and conduct experiments on a few toy data.   Most of the reviewers recognize the motivation, and effort of the paper.  During the discussion, the authors also successfully addressed some of the reviewers' questions.  However, concerns about the novelty as well as the lack of in-the-wild evaluation remain.   Reviewers acknowledge the problem of domain alignment using normalizing flows is hard, and this paper does not make any significant breakthroughs in this matter.  But, it seems it has made a small measurable step towards solving it, which is good enough to be recommended to accept.


**Award:**

No

---

### Decision · Program_Chairs · 2022-09-14

Accept